# Trade-offs in land-based carbon removal measures under 1.5 °C and 2 °C futures

Xin Zhao [1] ✉, Bryan K. Mignone [2], Marshall A. Wise [1] & Haewon C. McJeon[1,3]

Land-based carbon removals, specifically afforestation/reforestation and bioenergy with carbon capture and storage (BECCS), vary widely in 1.5 °C and 2 °C scenarios generated by integrated assessment models. Because underlying drivers are difficult to assess, we use a well-known integrated assessment model, GCAM, to demonstrate that land-based carbon removals are sensitive to the strength and scope of land-based mitigation policies. We find that while cumulative afforestation/reforestation and BECCS deployment are inversely related, they are both typically part of cost-effective mitigation pathways, with forestry options deployed earlier. While the $CO_2$ removal intensity (removal per unit land) of BECCS is typically higher than afforestation/reforestation over long time horizons, the BECCS removal intensity is sensitive to feedstock and technology choices whereas the afforestation/reforestation removal intensity is sensitive to land policy choices. Finally, we find a generally positive relationship between agricultural prices and removal effectiveness of land-based mitigation, suggesting that some trade-offs may be difficult to avoid.

The Intergovernmental Panel on Climate Change (IPCC) Sixth Assessment Report (AR6) estimates that the remaining carbon budgets for limiting global warming to well below 2 °C and 1.5 °C are 1150 and 500 Gigatons $CO_2$ ($GtCO_2$), respectively[1]. Land-based carbon dioxide removal (CDR) measures, especially afforestation/reforestation (A/R) and bioenergy with carbon capture and storage (BECCS), may be critical to achieving the long-term mitigation goals of the Paris Agreement[2-4]. In the IPCC AR6 scenario database[5], which compiles over 600 mitigation pathways generated by integrated assessment models (IAMs), the mean (*interquartile range*) cumulative land-based CDR between 2020 and 2100 is 460 (*350–560*) $GtCO_2$, including 100 (*10–190*) $GtCO_2$ from land use, land-use change and forestry (LULUCF) and 360 (*245–455*) $GtCO_2$ from BECCS (Fig. 1). While BECCS was deployed in almost all these pathways, about three-quarters of them also relied on net future LULUCF carbon removals, mainly through A/R, while other CDR methods were not extensively utilized (see Section S1 in Supplementary Information (SI)).

Both BECCS and land-system mitigation policies that incentivize carbon storage (e.g., A/R) could be land-intensive. BECCS relies on advanced technologies to convert lignocellulosic biomass, potentially sourced from purpose-grown energy crops, into modern energy carriers while capturing biogenic carbon and storing it underground in a geologic formation. In contrast, A/R entails expanding forests to enhance the carbon sequestered in vegetation and soil. Land-based policies, where they exist, have also focused on related activities, such as preventing deforestation and conserving natural ecosystems.

Land-based mitigation encourages regional resource competition and reallocation as land is physically limited and spatially heterogeneous[6]. In addition, the nonlinear trajectory of $CO_2$ uptake from land-based CDR measures can interact with other time-varying factors, such as agricultural productivity, technological progress, and socioeconomic drivers, introducing additional complexities[7,8]. Recent studies have explored the biophysical and economic mitigation potential of A/R[9-12] and BECCS[13-15]. However, when investigated independently, studies examining different CDR measures may overlook the trade-offs that arise from land competition under mitigation targets[16,17]. Capturing key differences among land-based CDR measures, including economic, environmental, and institutional differences, is crucial for evaluating their effectiveness and sustainability implications in long-term mitigation pathways.

[1]Joint Global Change Research Institute, Pacific Northwest National Laboratory, 5825 University Research Ct, College Park, MD, USA. [2]ExxonMobil Technology and Engineering Company, Annandale, NJ, USA. [3]KAIST Graduate School of Green Growth & Sustainability, Daejeon, Republic of Korea.
✉ e-mail: xin.zhao@pnnl.gov

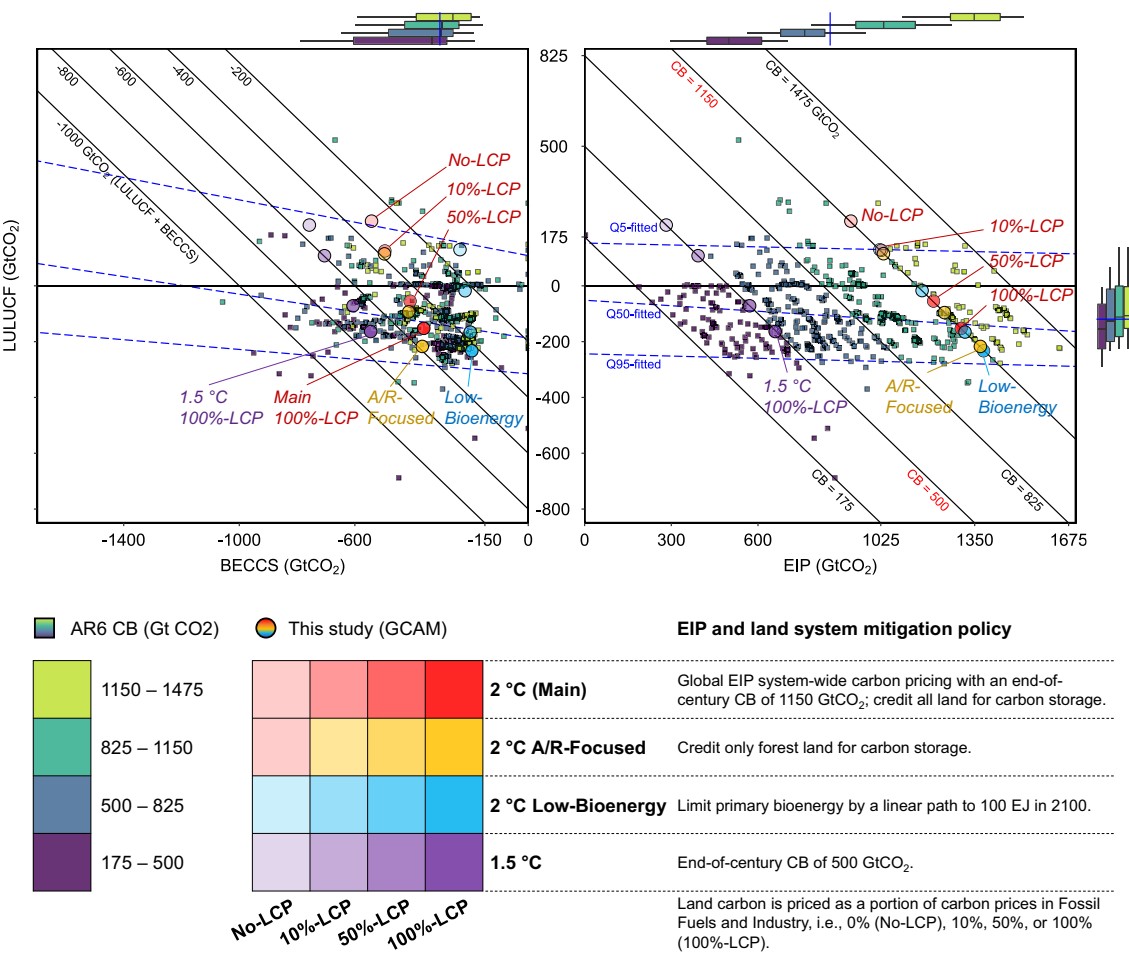

**Fig. 1 | Contributions of land-based carbon dioxide removal measures.** The left panel shows the relationship between global cumulative $CO_2$ removals/emissions in 2020–2100 for LULUCF and BECCS, while the right panel shows the relationship between LULUCF and Energy and Industrial Process (EIP) emissions, projected by climate change mitigation pathways. Each dot represents a projection from an IPCC AR6 pathway (square) or a mitigation pathway generated in the present study using GCAM (round). The square dots ($n = 604$) in both panels are projections from IPCC AR6 1.5 °C and 2 °C pathways and with Carbon Budgets (CBs) in [175, 1475] GtCO$_2$, distinguished by CB subranges (filled color). The boxplots on the sides show the median values (line), the 1st and 3rd quartiles (boxes), and the 5–95 percentile ranges (whiskers) of the AR6 pathways; the blue line on the boxplots shows the median value of the full range. The blue dotted lines in the main panels are fitted using quantile regression at the 5th, 50th, and 95th percentiles. The round dots ($n = 15$) represent GCAM projections, with scenarios distinguished by filled colors and described in the legend. Note that Main and A/R-Focused scenarios are identical under no land mitigation policy (No-LCP). Points on a diagonal line in a panel have the same total removals/emissions, i.e., land-based carbon removals (left) or CBs (right). The beta coefficient between LULUCF and BECCS, with CB controlled, stands at −0.17 for AR6 pathways. In contrast, the corresponding value for GCAM scenarios studied is −0.86 (−2 for the 2 °C Main scenarios). For more details about AR6 pathways, see Section S1. Data source: AR6 Scenario Database and GCAM simulation results.

Pooled data from AR6 suggests a weak yet statistically significant negative relationship between the cumulative removals from BECCS and LULUCF when the carbon budget is controlled (see fitted lines in the left panel of Fig. 1). At the median level, 16% of the emissions removal from a marginal increase in BECCS deployment could be leaked through an increase in LULUCF emissions. This leakage suggests a potential trade-off between BECCS and LULUCF removals as large-scale bioenergy deployments may induce land use change emissions[18,19]. Nevertheless, the implied emission leakage may be difficult to quantify precisely, given differences in modeling assumptions and policy implementation[20].

Recent IAM intercomparison efforts have highlighted the critical role of land-based CDR in long-term global decarbonization[21] and explored further implications for food security and climate overshoot[22,23]. Despite harmonized protocols, there is limited agreement among the models regarding the contribution of different land-based mitigation options[24]. Prior studies have broadly investigated BECCS-related assumptions[25,26]. However, we find a fundamental disparity among the models regarding whether and to what extent a land-system mitigation policy is considered[23]. For instance, some models protect forest land or incentivize A/R[27,28], while others apply a universal carbon tax (UCT) regime to value carbon stock in all land at the same price as carbon emissions from Fossil Fuels and Industry (FFI)[29,30]. Recent studies have not explored the sensitivity of land-based CDR deployment to the strength and scope of land-system mitigation policies, and the limited intra-model LULUCF variation in AR6 pathways (Fig. S4 in SI) makes it difficult to fully understand potential drivers and trade-offs in existing scenarios.

By comparing a UCT regime with a regime that includes mitigation efforts only in FFI sectors, studies have shown potentially substantial climate benefits[30,31], such as a lower cost of mitigation, arising from the inclusion of land-system mitigation policies. However, when land-based CDR is deployed on a large scale, sustainability concerns, including excessive resource use, biodiversity loss, and food insecurity, may arise[32–37]. In practice, due to sustainability concerns[22], potential biophysical impacts (e.g., albedo and evapotranspiration)[38], and

institutional challenges (e.g., measurement, reporting, verification, and permanence protocols)[39–42], the strength, scope, and effectiveness of land-system mitigation policies could be highly uncertain and, thus, deserves careful consideration by decision-makers. Compared to the UCT regime, forest protection or A/R policies, e.g., the expanded Reducing Emissions from Deforestation and Forest Degradation (REDD + ) framework or zero-deforestation supply chain policies, may represent land-system mitigation policies with only partial strength and/or partial land coverage, potentially limiting the amount of mitigation that can be delivered[43,44].

Here we explore how various mitigation policy choices affect the contribution of land-based CDR measures, their trade-offs, and the corresponding global market-mediated responses such as land use change, bioenergy supply, and agricultural price impacts. We employ a widely used global IAM, the Global Change Analysis Model (GCAM), to compare systematically designed scenarios with different strengths and sector coverages of land-based mitigation efforts (see Fig. 1 and "Methods"). The main scenarios (2 °C) limit the end-of-century carbon budget to 1150 $GtCO_2$ by applying a global energy system carbon price with different strengths of land carbon pricing (LCP) applied to all land. The No-LCP scenario assumes no land mitigation policy, whereas the 100%-LCP scenario applies the same price to land carbon storage and FFI carbon. These two scenarios correspond to the FFI carbon tax and UCT scenarios studied by Wise et al.[30] but reflect numerous model improvements since that time. To address the uncertainty surrounding land-system policies and to examine the linearity of the response, we add two partial land carbon pricing cases with different strengths, i.e., 10%-LCP and 50%-LCP. Given the scope of the LCP policies, land-system mitigation can be broadly defined as removals through LULUCF, and it also encompasses actions aimed at alleviating or preventing deforestation and natural land depletion.

Relative to the main scenarios, we also study two sets of cases with carbon pricing applied asymmetrically within a sector: (1) pricing only forest carbon (asymmetric carbon pricing within the land sector) with no mitigation policy on non-forest land (A/R-Focused), and (2) lower carbon credit to BECCS (asymmetric carbon pricing within the energy sector) by limiting primary lignocellulosic bioenergy to 100 EJ per year by 2100 (Low-Bioenergy). The policy scope of land-based mitigation is among the most overlooked areas in the IAM literature. Therefore, a comparison between the Main scenarios and the A/R-Focused scenarios offers valuable insights into differences in policy effectiveness and implications for agricultural commodity prices. In GCAM, the Low-Bioenergy scenario effectively lowers the shadow price of carbon used to provide incentives for BECCS. In addition, another set of scenarios (1.5 °C) that limit the end-of-century carbon budget to 500 $GtCO_2$ are also added as the climate target is the most widely investigated dimension in the literature. More detailed information regarding the modeling and scenarios is available in "Methods" and Section S2.

## Results

### Trade-offs between land-based mitigation measures

Outcomes from our mitigation scenarios with regard to LULUCF and BECCS removals encompass a large share of the variability observed in AR6 pathways (Fig. 1 left panel). The 2 °C Main scenario with all land system carbon fully priced (100%-LCP) projects a total of 510 $GtCO_2$ land-based CDR, including 360 $GtCO_2$ from BECCS and 150 $GtCO_2$ from LULUCF. If the land mitigation policy is weaker, the net LULUCF removal weakens and could become an emission source (230 $GtCO_2$ in No-LCP), while the contribution of BECCS increases (540 $GtCO_2$ in No-LCP). For a given carbon budget, this suggests a clear trade-off between LULUCF and BECCS when the strength of land mitigation policy is varied, driven by resource competition. The 2 °C A/R-Focused scenarios, which specifically support forest carbon storage, result in higher total land-based CDR compared to the Main scenarios (e.g., by 70 $GtCO_2$ in 100%-LCP). Limiting bioenergy in Low-Bioenergy

scenarios reduces the deployment of BECCS by more than 50% to 190–230 $GtCO_2$ compared to the Main scenarios (360–540 $GtCO_2$). Only about half or less of the reduction in BECCS is compensated by higher net LULUCF removals, resulting in lower total land-based CDR. In addition, with a more stringent climate target (i.e., lower carbon budget by 650 $GtCO_2$ in 1.5 °C relative to 2 °C scenarios), removals from land-based CDR increase by 200–230 $GtCO_2$ compared to the Main scenarios. Most (over 90%) of the additional removal is from BECCS, echoing previous findings that BECCS is more responsive to climate targets[23].

With a stronger land mitigation policy, the increase in LULUCF net removal outweighs the decreased removal from BECCS, leading to a higher total land-based CDR. However, the marginal effectiveness of the policy (in terms of $CO_2$ removal) decreases as the strength of the land carbon price increases. For instance, in 2 °C Main, the total land-based CDR is 60 $GtCO_2$ higher in 10%-LCP compared to No-LCP, whereas it is only 45 $GtCO_2$ higher in 100%-LCP compared to 50%-LCP. The nonlinear responses to variation in LCP strength are similar in different sets of policy scenarios, and they indicate that imposing a weak land mitigation policy might have a more pronounced impact than strengthening an existing policy and that avoiding deforestation could be more effective than A/R. Collectively, our scenarios reveal a more pronounced trade-off between LULUCF and BECCS compared to what is suggested by AR6 pathways, with a beta coefficient of −0.86 (in contrast to −0.17 in AR6). This supports our hypothesis that variations in LULUCF across AR6 pathways are predominantly driven by uncontrolled model and scenario differences rather than land-system mitigation policy choices.

The FFI carbon prices in our scenarios, ranging from $33 to $83 in 2025, align generally with the range of $10–$122 (10th–90th percentile) estimated in AR6 pathways for 2025 (Fig. 2). However, as suggested by our results, different policy choices may result in varying shadow prices of carbon on land-based CDR options as well as different FFI carbon prices. Our results demonstrate that FFI carbon prices in 100%-LCP scenarios are about 16% (2 °C Main) to 30% (Low-Bioenergy) lower than those in No-LCP scenarios. Compared to the Main scenarios, depending on the strength of land policies, FFI carbon prices could be up to 10% lower in A/R-Focused scenarios, 25–50% higher in Low-Bioenergy scenarios, and 80–90% higher in 1.5 °C scenarios.

### Connecting land use and carbon removal

As land is a key input for producing carbon mitigation or storage services, it is imperative to understand how land is used and the corresponding productivity in terms of carbon removal (i.e., removal intensity). Here, we illustrate the connection between carbon removal and the corresponding land use, focusing on a Main scenario (2 °C Main & 100%-LCP) to set the foundation for subsequent scenario comparisons. In 2 °C Main & 100%-LCP, the mean annual land-based CDR is 6.3 $GtCO_2$ $yr^{-1}$, with 4.4 $GtCO_2$ $yr^{-1}$ from BECCS and 1.9 $GtCO_2$ $yr^{-1}$ from LULUCF (Fig. 3A). Of the BECCS, 60% (2.6 $GtCO_2$ $yr^{-1}$ or 213 $GtCO_2$) relies on purpose-grown energy crops, and the remaining 40% (1.8 $GtCO_2$ $yr^{-1}$ or 145 $GtCO_2$) uses crop and forestry residues and municipal solid wastes (MSWs) as feedstock, with negligible direct land implications. Global net-zero carbon emissions are achieved in 2085, at which point FFI emissions of 10.3 $GtCO_2$ $yr^{-1}$ are offset by land-based CDR (Fig. 3B). On average over the study period, about 120 exajoules per year (EJ $yr^{-1}$) of primary modern bioenergy is produced, corresponding to 10 $GtCO_2$ $yr^{-1}$ in biogenic carbon (Fig. 3C). However, only about 44% of the biogenic carbon is captured using BECCS since 67% of the primary bioenergy is used in technologies combined with CCS in which, on average, 66% of the primary biomass carbon is sequestered and stored (Figs. S19–S21).

About 65 EJ $yr^{-1}$ of the primary bioenergy, or 5.5 $GtCO_2$ $yr^{-1}$ in biogenic carbon, is supplied by purpose-grown energy crops, requiring, on average, about 300 Mha of dedicated energy cropland

(Fig. 3D). In addition to energy cropland expansion, as land carbon is fully priced, about 180 Mha is demanded globally for A/R, along with 16 Mha other natural land restoration. Over 72% of the global land required for BECCS and A/R comes from converting pasture (360 Mha; mostly unmanaged), whereas the remainder comes from converting non-energy cropland (28% or 140 Mha), which leads to higher agricultural prices. Notably, the majority (95%) of BECCS removals occur in the second half of the century, while for LULUCF, this share is about 48%. The temporal pattern of land-based CDR is a co-evolution of several key factors, such as primary biomass supply, CCS technology deployment, and other techno-economic assumptions. Specifically, the temporal pattern, when land carbon is fully priced, reflects the relative ease with which land can be converted to forest on the one hand and the slower deployment of a more capital-intensive, emerging technology in the energy system (e.g., BECCS) on the other.

The aggregate land removal intensity, attributing all land-based CDR to the use of energy cropland and forest land, is 13.1 $tCO_2ha^{-1}yr^{-1}$ (Fig. 3E). Our decomposition shows that the removal intensity is higher for BECCS than LULUCF (14.6 vs.10.4 $tCO_2ha^{-1}yr^{-1}$). However, the removal intensity for BECCS would be 40% lower (8.7 $tCO_2ha^{-1}yr^{-1}$) if not accounting for waste & residue-based BECCS, and then 36% higher (11.8 $tCO_2ha^{-1}yr^{-1}$) if further disassociating land that is used for bioenergy without CCS (Fig. 3F). On the other hand, the net land carbon stock in afforested areas stands at 193 $GtCO_2$, implying a marginal forest carbon density of 13.1 $tCO_2ha^{-1}yr^{-1}$, and 23 $GtCO_2$ in other natural land (Table S5). However, only the additional carbon storage matters, which explains the relatively lower LULUCF removal intensity. The full set of mitigation pathway results that connect land-based CDR to implications for land use competition is provided in Figs. S13–S27 and Table S5 and discussed in Section S3.2.

## Impact of land carbon pricing strength

The strength of land carbon pricing has a considerable impact on the breakdown of carbon budgets and their corresponding land use implications (Fig. 3). A stronger land mitigation policy places greater weight on carbon storage in determining land uses, resulting in increased demand for forest and other natural land and reduced land use for food

and energy. Conversely, a weaker land mitigation policy would result in the opposite effect. Compared to 100%-LCP, forest and other natural land use is, on average, 115 Mha lower in 50%-LCP and 445 Mha lower in No-LCP, while cropland use is higher by about 100 Mha in 50%-LCP and 355 Mha in No-LCP, split almost evenly between energy and non-energy crops. The increased energy cropland leads to a higher primary bioenergy supply from energy crops, e.g., 10 $EJ yr^{-1}$ higher in 50%-LCP and 32 $EJ yr^{-1}$ higher in No-LCP compared to 100%-LCP. In contrast, waste & residue-based primary bioenergy is not substantially affected.

Global non-energy cropland expands over time in the reference scenario (Section S3.1), but this trend is weakened when Energy and Industrial Processes (EIP) carbon is priced in the 2 °C No-LCP scenario as the demand for energy crops is higher and reversed with stronger land carbon policies as land competition further intensifies due to the higher demand for A/R and natural land (Fig. 3D). Global deforestation switches to afforestation with a land policy in-between 10%-LCP and 50%-LCP. The impact of varying LCP on land use patterns and primary bioenergy supply is also consistent at the regional scale (Figs. S18 and S24).

The aggerate land removal intensity is smaller with weaker LCPs, e.g., 11.2 $tCO_2ha^{-1}yr^{-1}$ in No-LCP. For BECCS, when the land carbon policy is weaker, BECCS demand is higher, the transition of biomass to sectors in which it can be used with CCS is faster, and the overall capture rate is higher, while the share of biomass sourced from energy crops is also higher and crop yields are lower as land competition is weaker (Fig. 3F). Our decomposition highlights that (1) BECCS produced using wastes and residues plays a key role in improving the aggregate removal intensity and is less sensitive to land carbon pricing strength, (2) the trade-off in land-based mitigation measures is primarily between energy crop-based BECCS and LULUCF, driven by land competition, (3) while uncertain, the relatively larger (more negative) LULUCF removal intensity in net deforestation scenarios (No-LCP & 10%-LCP) reveals that the rate of LULUCF emissions from deforestation surpasses the rate of sequestration in A/R, and (4) stronger land mitigation policies encourage land allocation decisions that reflect potential changes in land carbon sequestration, resulting in a more balanced land use between energy crops and forest and higher aggregate removal intensity.

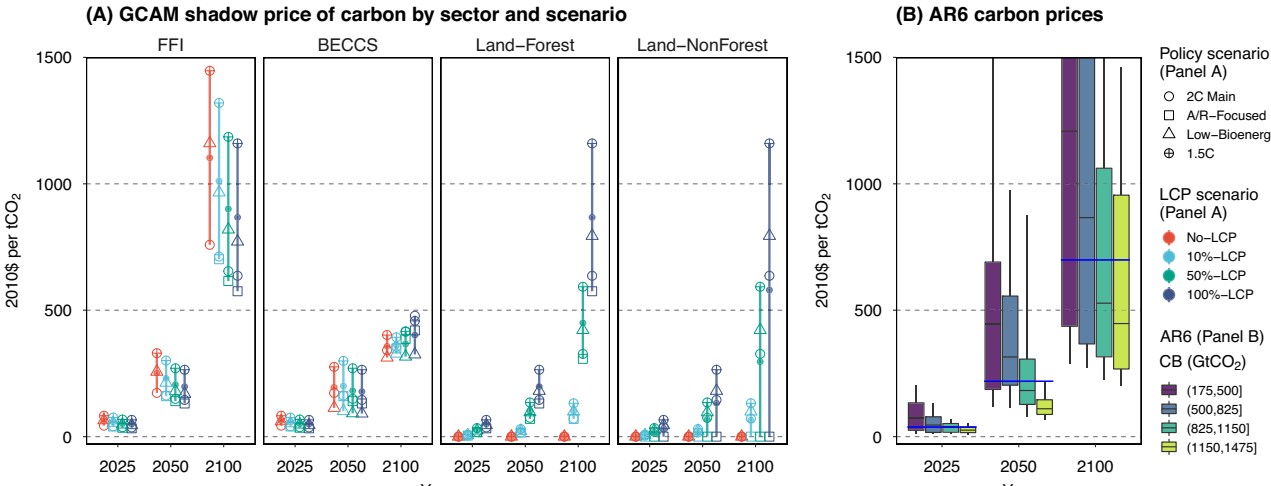

**Fig. 2 | Carbon prices in mitigation pathways.** Panel (**A**) displays the shadow price of carbon by sector across pathways investigated in this study, with each dot representing the projected carbon price in a study year and sector, distinguished by policy scenario (point shape) and land-system carbon pricing (LCP) scenario (color). The corresponding point-range per LCP scenario group is added with a line indicating the range of the dots and a solid dot indicating the mean value. Note that the difference in the shadow price of carbon across sectors reflects various mitigation policy choices. Panel (**B**) shows the carbon price distributions across AR6

pathways (n = 565) by year and Carbon Budget (CB) subranges. The boxplots show the median values (horizontal line within the boxes), interquartile range (boxes), and the 10th–90th percentile ranges (whiskers) of the AR6 pathways (truncated at a maximum of $1500 per $tCO_2$); the blue line on the boxplots shows the median value of the full CB range. Additional information is provided in Fig. S13 and Tables S3 and S4. Data source: GCAM simulation results and AR6 Scenario Database.

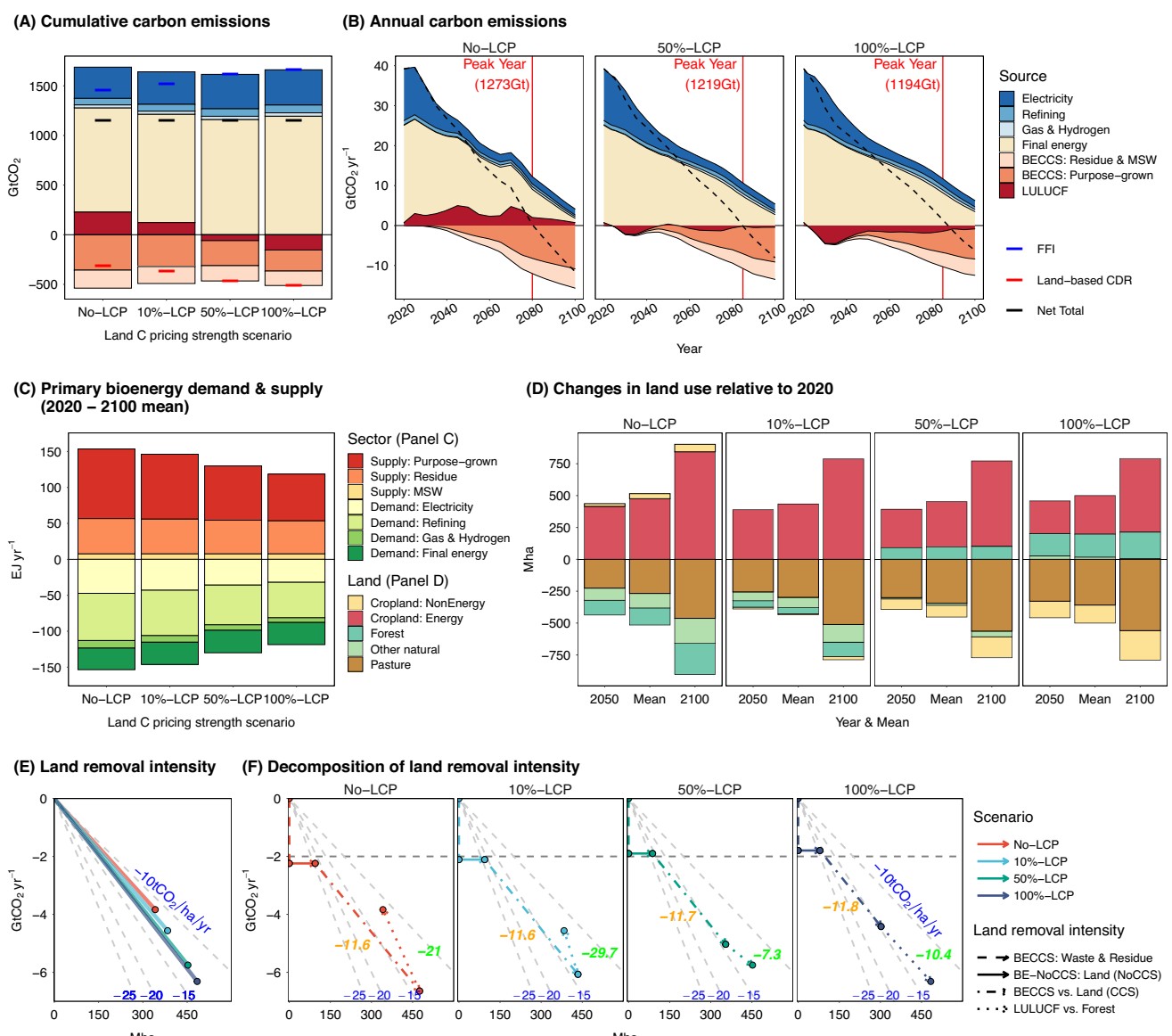

**Fig. 3 | Impact of land carbon pricing strength on key outcomes under 2 °C scenarios.** Projections from the Main (2 °C) scenarios with varying land carbon pricing strengths, from no land carbon pricing (No-LCP) to 100% land carbon pricing (100%-LCP), are presented for global cumulative (**A**) and annual (**B**) carbon emissions, primary bioenergy demand and supply (**C**), land use change (**D**), land removal intensity (**E**), and the decomposition of land removal intensity (**F**). All scenarios have a net total cumulative emissions of 1150 GtCO₂. Carbon dioxide emissions/removals in (**A**) and (**B**) show contributions by Fossil Fuels and Industry (FFI), BECCS (by feedstock sources: residue & MSW or purpose-grown energy crop), and LULUCF. The final energy sectors include industry, buildings, and transportation. The peak carbon emission year and corresponding cumulative emissions are highlighted (red vertical lines) in (**B**). Stacked bars in (**C**) show the 2020–2100 mean primary second-generation bioenergy supply by source (positive values) and demand by sector (negative values). Stacked bars in panel (**D**) present land use change decomposition by 2050 and 2100 and the mean value in 2020–2100. Points in (**E**) present the relationship between the 2020–2100 mean land-based CDR and the corresponding land utilized so that the slope of the lines is a measure of the 2020–2100 mean land removal intensity. The slope of background lines (gray), for reference, is labeled (in blue with a unit of tCO₂ per hectare per year). Panel (**F**) decomposes panel (**E**) by land-based CDR sources and the corresponding land use, including waste & residue-based BECCS (vertical lines; no land attribution), energy cropland for bioenergy used in sectors without CCS (horizontal lines; no CCS attribution), energy crop-based BECCS vs. energy cropland for bioenergy used in CCS sectors (slopes annotated in orange), and LULUCF vs. forest (slopes annotated in green). Data source: GCAM simulation results.

Furthermore, different temporal patterns in BECCS and A/R are not entirely apparent from the aggregate trade-off between them, with more LULUCF removals occurring early in the century and more BECCS removals occurring later in the century. Consequently, stronger LCPs also reduce the magnitude of carbon budget overshooting, e.g., 123 (No-LCP) vs. 54 (100%-LCP) GtCO₂, and encourage higher total land-based CDR. The findings regarding the impact of land carbon pricing strength in the 2 °C Main scenarios are also consistent with the results observed in our other sets of scenarios.

## Sensitivity of CDR deployment to the scope of land and energy system policies

We further examine the sensitivity of CDR deployment to other policy choices, including the extent of policy coverage on land, constraints on primary bioenergy, and the choice of climate target, providing results for the 100%-LCP scenarios in Fig. 4. Specifically, when land mitigation policy only subsidizes carbon storage in forest (A/R-focused), compared to the Main scenario, other natural land becomes a major source of land (−315 Mha) along with more pasture conversion (−245 Mha) as

they are not credited for carbon storage anymore. As a result, considerably higher demand for forest land (650 Mha, on average) is seen (Fig. 4D), more than three times higher than the value in the Main scenario (180 Mha), along with moderately more expansion of energy cropland (+40 Mha) and less reduction in non-energy cropland (+50 Mha). The altered land use pattern by A/R has a relatively minor impact on primary bioenergy supply and BECCS removal but considerably increases the LULUCF removal by over 40% (+0.8 GtCO$_2$ yr$^{-1}$; Fig. 4A, C). LULUCF removal is relatively stable in A/R-Focused scenarios throughout the century as forest land continues to expand over time (Fig. 4B). The marginal forest carbon density increases by 15% compared to the Main scenario, but the leakage is larger since more non-forest land with relatively high carbon density is displaced, which also leads to higher initial LULUCF emissions compared to other scenarios. As a result, the overall forest removal intensity becomes considerably smaller (−60% to 4.2 tCO$_2$ha$^{-1}$yr$^{-1}$ compared to Main; Fig. 4F).

Under the Low-Bioenergy constraint, the supply of primary bioenergy decreases by about 45% to 66 EJ yr$^{-1}$, compared to the Main scenario. The linear constraint has a stronger impact on purpose-grown biomass compared to MSW & residue, as the former is mainly utilized later in the century. The mean purpose-grown biomass supply decreases by 66% to 22 EJ yr$^{-1}$, accompanied by a 64% reduction in energy cropland use. The corresponding BECCS CDR experiences an even larger decrease of 69% (−1.8 GtCO$_2$ yr$^{-1}$) due to the slower transition of biomass to sectors in which it can be used with CCS technologies (e.g., 8 percentage points lower in the share of biomass used in conjunction with CCS). The constraint, effectively resulting in weaker incentives for BECCS, dampens the price transmission from the carbon market to the primary biomass market. Although the carbon capture rate in BECCS is slightly higher, driven by the higher carbon prices, the relatively lower prices of primary biomass discourage crop yield intensification (−7%) as well as the transition of biomass use for BECCS. As a result, BECCS deployment is smaller, and energy cropland removal becomes less efficient, with the removal intensity (not including waste & residue) decreasing to 7.4 from 8.7 tCO$_2$ha$^{-1}$yr$^{-1}$ (Main), and to 11.1 from 11.8 (Main) when

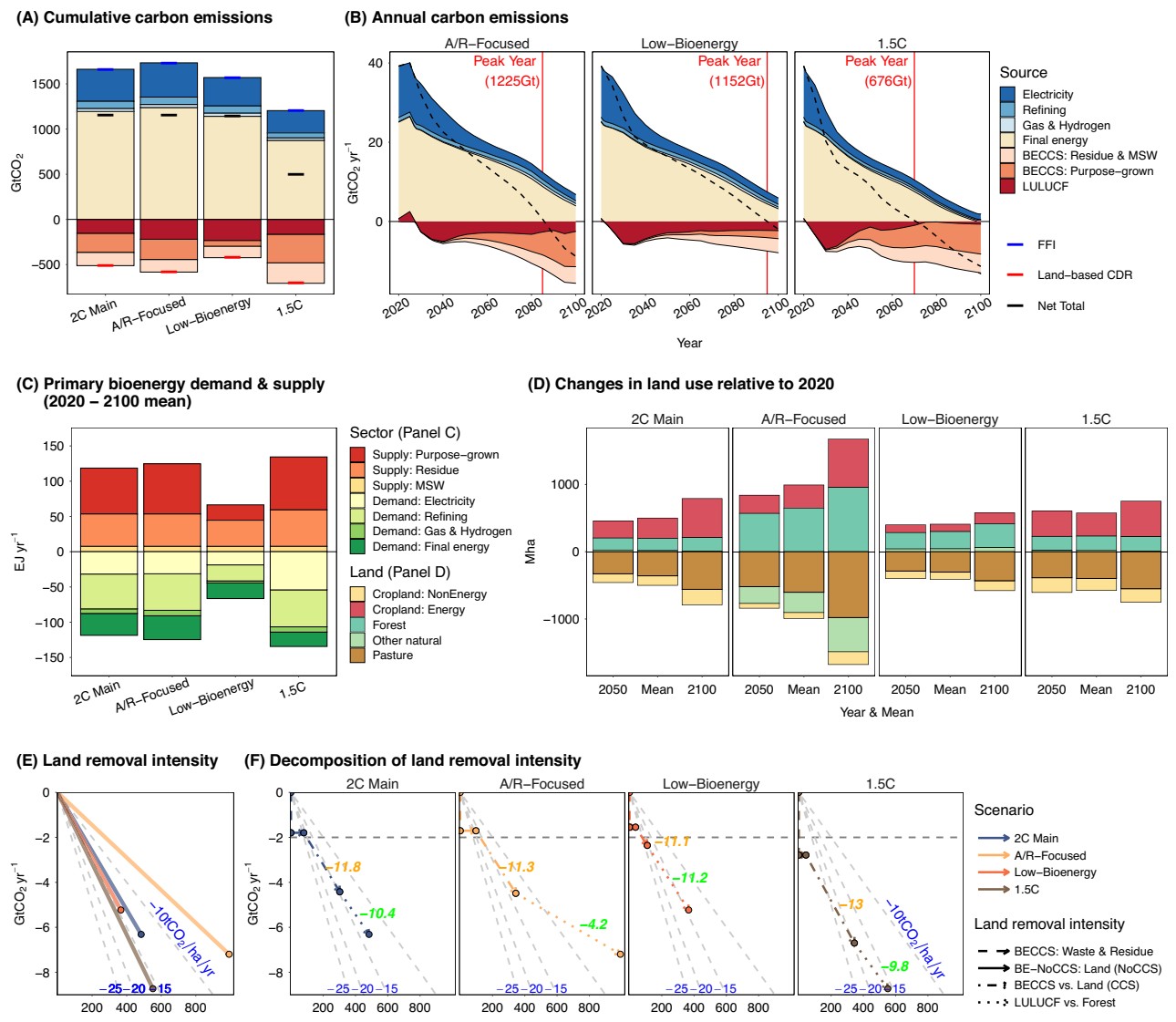

**Fig. 4 | Impact of alternative energy and land system policies on key outcomes under 100% land carbon pricing.** Projections from scenarios with alternative energy and land system policies, including 2 °C (main), A/R-Focused, Low-Bioenergy, and 1.5 °C, are presented for global cumulative (**A**) and annual (**B**) carbon emissions, primary bioenergy demand and supply (**C**), land use change (**D**), land removal intensity (**E**), and the decomposition of land removal intensity (**F**). All scenarios have 100% land carbon pricing (100%-LCP) in which land carbon is priced at the Fossil Fuels and Industry (FFI) carbon prices. See Fig. 3 caption for detailed panel descriptions. Data source: GCAM simulation results.

energy cropland that is attributed to biomass used without CCS is removed from the intensity calculation. The reduced land pressure from BECCS also yields efficiency benefits for A/R on the extensive margin, i.e., marginal forest carbon density increases (+ 9% compared to Main). Thus, higher land use for A/R (+75 Mha) is also seen, accompanied by more pronounced LULUCF removal (+ 1 GtCO$_2$ yr$^{-1}$) and, consequently, higher LULUCF removal intensity (+8% to 11.2 tCO$_2$ha$^{-1}$yr$^{-1}$), compared to the Main scenario.

Under a more stringent climate target (1.5 °C), an additional reduction of 2.4 GtCO$_2$ yr$^{-1}$ comes from land-based CDR, with the majority coming from BECCS (1.3 GtCO$_2$ yr$^{-1}$ from energy crops and 1 GtCO$_2$ yr$^{-1}$ from wastes and residues) and a smaller contribution from LULUCF (0.1 GtCO$_2$ yr$^{-1}$). This entails considerably higher carbon prices (i.e., +83%), driving a more carbon-efficient use of land, mainly for energy crop-based BECCS (i.e., +31% in removal intensity to 11.4 tCO$_2$ha$^{-1}$yr$^{-1}$). Higher carbon prices promote a faster transition of biomass use from sectors or technologies with no CCS (e.g., final energy sectors) or low carbon capture rates (e.g., ~50% in fuel refining) to those with higher capture rates (e.g., ~90% in electricity). As a result, 59% of the biogenic carbon in primary biomass is sequestered, which is considerably higher than the share in the 2 °C scenario (44%). The earlier deployment of BECCS also drastically increases the cumulative sequestration, echoing Obersteiner et al.[45] The primary bioenergy increase is higher in early periods, e.g., +21% before 2050 vs. +12% after 2050. The earlier deployment of BECCS and greater share of bioenergy used for BECCS overall explain how the 13% (+ 15 EJ yr$^{-1}$) average increase in primary bioenergy supply drives up BECCS removal by 52%.

With a moderate yield intensification, the increase in energy cropland use is +42 Mha, higher than the forest land use increase (+26 Mha), in the 1.5 °C scenario compared to the 2 °C scenario. In stark contrast to the substantially higher BECCS removal intensities, LULUCF removal intensity decreases by 6% to 9.8 tCO$_2$ha$^{-1}$yr$^{-1}$. This decline is mainly driven by extensive margin responses[46,47], meaning that productivity (for carbon removal) diminishes when land expands, especially under increased land competition. In the 2 °C scenario, the effective carbon stock in the expanded forest stands at 290 tonnes of carbon per hectare (tC/ha) for the full study period, about 30% higher than the median forest carbon density of 224 tC/ha (Fig. S7). However, amidst stronger land competition under a more stringent climate target (1.5 °C), the effective carbon stock in the expanded forest decreased by 7% to 270 tC/ha when the forest further expanded (+26 Mha), resulting in a lower marginal forest carbon density (−7% to 12.2 tCO$_2$ha$^{-1}$yr$^{-1}$). This response is not obvious for energy cropland, as it is offset by price-induced yield intensification.

## Land carbon removal intensity over time

Across our scenarios, the average removal intensity of land-based CDR ranges from 7.3 to 18.1 tCO$_2$ha$^{-1}$yr$^{-1}$ and from 4.7 to 10.8 tCO$_2$ha$^{-1}$yr$^{-1}$ when not accounting for waste & residue-based BECCS, over the full study period. The removal intensity of energy crop-based BECCS improves over time in our scenarios, with the scenario average for the full study period more than double the early period (2020–2050) value, i.e., 9.3 vs. 3.8 tCO$_2$ha$^{-1}$yr$^{-1}$. This improvement is driven by factors such as yield growth, sectoral transition, and technological progress. Particularly noteworthy is the rapid growth in the share of bioenergy used for BECCS over the first several decades (Fig. S19). Disregarding the sectoral transition effect by removing bioenergy not used with CCS, the increase over time in the intensity would be smaller, i.e., 11.9 (full period) vs. 11 (early period) tCO$_2$ha$^{-1}$yr$^{-1}$. On the other hand, the removal intensity of A/R is generally anticipated to decline in the long term, since A/R occurs in early periods and the marginal removal intensity declines as the forest matures. This decline in the average removal intensity of LULUCF is observed in our runs, with the value over the full study period 42% lower than the value over the early period (2020–2050) in the scenarios with A/R.

The land carbon removal intensity of BECCS and LULUCF in our scenarios largely span the AR6 ranges, despite the substantial variation in both sources (Fig. 5 and Section S3.3). In the full study period, in AR6 pathways, the median removal intensity of energy crop-based BECCS is 9.5 tCO$_2$ha$^{-1}$yr$^{-1}$, which is considerably higher than the LULUCF removal intensity of 5.7 tCO$_2$ha$^{-1}$yr$^{-1}$. This finding is consistent with our results, although the difference between the two types of land-based CDR is relatively smaller in our scenarios, 9.6 vs. 7.6 tCO$_2$ha$^{-1}$yr$^{-1}$. However, the median LULUCF removal intensity in the AR6 scenarios

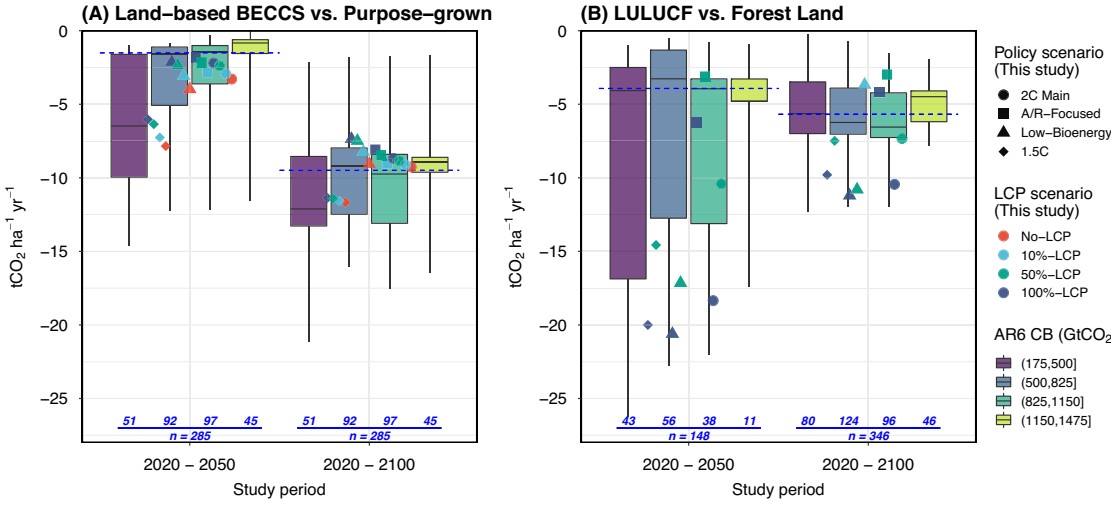

**Fig. 5 | Land carbon removal intensity.** Panels (**A**) and (**B**) show land carbon removal intensity for energy crop-based BECCS versus purpose-grown energy cropland and LULUCF versus forest land, respectively. The dots in each panel represent mean projections from the present study (GCAM) by study period (2020–2050 or 2020–2100), policy scenario (point shape), and land-system carbon pricing (LCP) scenario (color). The boxplots exhibit the distribution of results across AR6 pathways, including the median (horizontal line within the boxes), interquartile range (boxes), and 5–95 percentile range (whiskers) by study period and Carbon Budget (CB) subrange. The blue dotted line represents the median value in the full CB range for the available AR6 pathways. Note that only a subset of AR6 pathways reported quality land projections. We cannot further decompose BECCS removal intensity for AR6 pathways to associate purpose-grown cropland with CCS sectors due to inadequate and/or low-quality reporting of the relevant data needed. In Panel (**B**), pathways or study periods with net global deforestation were removed for meaningful comparisons of the removal intensity, and negative values represent net removal in the study period. The number of AR6 pathways by CB subrange is annotated in blue. Data sources: GCAM simulation results and AR6 Scenario Database.

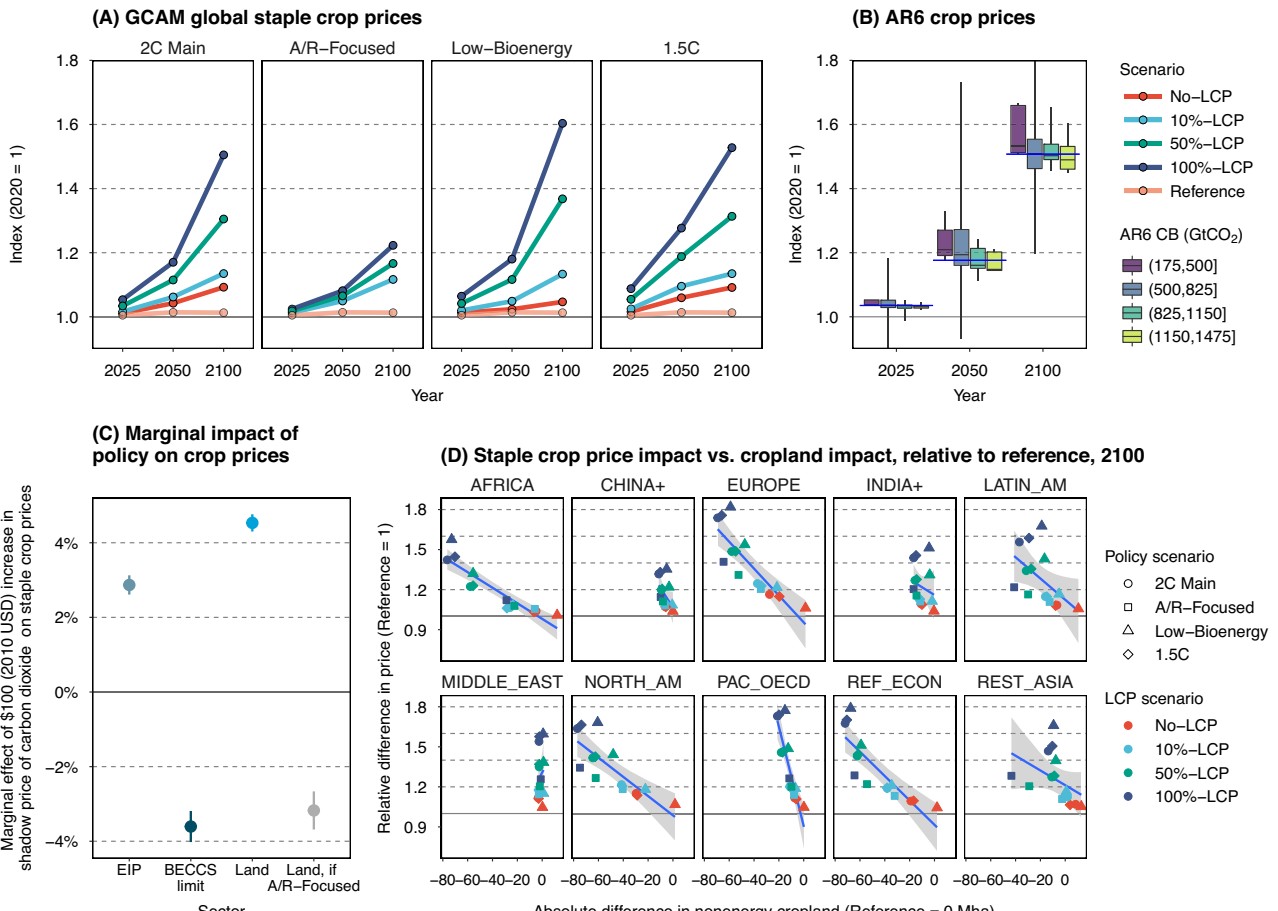

**Fig. 6 | Implications of land-based mitigation on crop prices.** Panel (**A**) displays the staple crop price index (2020 = 1) in pathways explored in this study, with each dot representing the crop price in a projection year (i.e., 2025, 2050, 2100), distinguished by policy scenarios (subpanels), and scenarios (lines & colors). Staple crop in GCAM is an aggregation of wheat, corn, rice, other grains, and root & tuber (Table S2). Panel (**B**) shows the crop price index (2020 = 1) distributions across AR6 pathways (n = 28) and crops (n <= 4) by year and Carbon Budget (CB) subranges. AR6 crops include wheat, corn, rice, and soybean. The boxplots show the median (horizontal line within the boxes), interquartile range (boxes), and 5–95 percentile range (whiskers) by study period and CB subrange. The blue line represents the median value in the full CB range. Panel (**C**) shows the marginal effect on staple crop prices (dots) from a $100 (2010 USD) increase in the shadow price of carbon dioxide by sector. The coefficients correspond to Model 4 in Table S7. Error bars represent the 95% confidence intervals (CI) of each coefficient. Panel (**D**) shows the relationship (point) between staple crop price impact (reference = 1) and cropland impact (reference = 0 Mha) in 2100 across LCP scenarios (point colors) and policy scenarios (point shapes) by AR6 R10 regions (subpanels; see Table S1 for mappings). Linear trend lines (blue lines) and 95% CI (gray ribbons) are added for each region (subpanel). Data source: GCAM simulation results and AR6 Scenario Database.

increases between the earlier period (2020–2050) and the full study period, which could reflect differences in the timing of A/R deployment as well as differences in policy scope and the overall LULUCF composition. The substantial uncertainty in land removal intensity underscores the importance of conducting thorough model intercomparison studies to not only explore the variations in results but also investigate the underlying differences in data and parameter assumptions. In addition, one should be cautious when comparing results between IAM and sectoral analyses[4,9,11,48] due to potential discrepancies in metrics and categorization. For instance, if BECCS based on waste and residue is not distinguished, the estimated BECCS land removal intensity could be considerably different.

### Agricultural price implications of land-based mitigation measures

World average staple crop prices vary considerably across the mitigation scenarios studied, ranging from +2% (Low-Bioenergy & No-LCP) to +28% (1.5 C & 100%-LCP) by 2050 and from a + 5% increase (Low-Bioenergy & No-LCP) to a + 60% increase (Low-Bioenergy & 100%-LCP) by 2100 (Fig. 6A). In contrast, reference prices are relatively stable over

the century (+1%). The impacts on agricultural prices at regional and sectoral levels are heterogeneous and may be more substantial (Fig. S33). Scenarios with stronger land carbon policies, such as 50%-LCP and 100%-LCP, align more closely with AR6 pathways, although agricultural prices are among the least reported variables in those pathways (Fig. 6B). The large uncertainty in price transmission from the carbon market to agricultural markets has been highlighted in recent studies[36,49].

Due to the variation in the strength and scope of land-system mitigation policies in our scenarios, we are able to conduct a deeper investigation into crop price responses ("Methods"). Our findings demonstrate that a $100 per tCO$_2$ increase in carbon prices when implemented across all sectors, results in a 7.4% rise in staple crop prices. Of this increase, 2.9% is attributed to carbon pricing in the EIP sector, while 4.5% is driven by carbon pricing in the land system (Fig. 6C). The effect attributed to EIP is further partitioned into a −0.7% decrease from FFI and a + 3.6% increase from BECCS. Stronger mitigation of fossil fuel emissions, all else equal, reduces the burden on land, resulting in a negative agricultural price impact. Although it puts pressure on land, BECCS has a relatively smaller impact on crop

prices than direct land carbon pricing due to its higher long-term removal intensity and ability to use nonland-based feedstocks, such as wastes and residues. However, when land pricing is restricted to forest land, the effect of land carbon pricing on crop prices is reduced to +1.4% as the greater conversion of unmanaged pasture and other natural lands alleviates the pressure on cropland. While the magnitude of the price transmission can be sensitive to several model parameters, our findings suggest that differentiating carbon prices by sector substantially enhances the explanation of their price transmission to agricultural markets (e.g., Models 1–4 vs. Model 8 in Table S7). This distinction is crucial as relying solely on FFI or EIP carbon prices to explain land use implications could lead to mis-interpretation of the results.

Stronger land competition with non-energy cropland, driven by land-based mitigation measures, leads to higher agricultural prices, as implied by the negative correlation between the two, particularly at the regional level (Fig. 6D). Notably, price-induced yield intensifications and demand-side adaptations can substantially mitigate consumer impacts. For instance, in a scenario with high price impacts, i.e., +50% by 2100 in Main & 100%-LCP, the global staple cropland area decreased by 24% (or 170 Mha) compared to the reference scenario. However, staple crop production only decreased by 5% (or 300 Mt) as yield increased by 25%, mainly leading to lower feed consumption. The market-mediated responses are also more robust when land mitigation policies are stronger (Figs. S34–S36).

## Discussion

Recent studies have highlighted the substantial uncertainty in pro-jecting land-based CDR measures and their economic and environ-mental implications[4,22–24,49,50]. To further clarify and assess the role of land-based CDR, we examined the response to key assumptions underlying the representation of BECCS and land-system mitigation options in GCAM and explored mitigation pathways with different land-based mitigation policies. By exploring and comparing these scenarios within an internally consistent framework, we demonstrated that the amount and type of CDR, as well as other implications of CDR deployment, are sensitive to the strength and scope of land-based mitigation policies.

Across our scenarios, the end-of-century cumulative land-based CDR ranged from 100 to 700 GtCO₂. In the absence of any land-system mitigation policy, the deployment of BECCS was the highest (230–540 GtCO₂), which also encouraged higher emissions from LULUCF, ran-ging from 130 to 230 GtCO₂ compared to 110 GtCO₂ in the reference scenario. Valuing land carbon storage, even partially, reduced emis-sions leakage from energy crop-based BECCS and lowered the cost of mitigation in the energy system. With full land pricing, the land system provided a net carbon sink between 150 and 230 GtCO₂, reducing the deployment of BECCS compared to the case without land pricing but increasing total land-based CDR. Our results suggested an inverse relationship (stronger than implied by AR6 pathways) between cumulative BECCS and LULUCF removals in mitigation scenarios, reflecting land competition.

Upon a thorough comparison of the scenarios, we highlight sev-eral key insights relevant to decision-makers. First, BECCS and A/R both contribute substantially to mitigation in 1.5 °C and 2 °C scenarios. Our scenarios suggest that BECCS and land-system mitigation mea-sures are both typically part of cost-effective mitigation pathways. Although the cumulative deployment of BECCS and A/R are found to be inversely related, reflecting land competition, we also find that their deployment is separated in time, with A/R deployment occurring earlier, as a less expensive option, and BECCS occurring later as technologies and supply chains develop and mature.

Second, the removal intensity of BECCS is typically higher than A/R over long time horizons. In general, we find that the amount of carbon removal per land utilized (the removal intensity) is higher for BECCS than for A/R in aggregate. A given amount of land set aside for bioenergy can enable biomass production indefinitely, which, when coupled with CCS, provides a steady stream of removals. In contrast, a given amount of land used for A/R will lead to larger carbon storage initially but will diminish over time as the forest matures. This implies that the removal intensities are sensitive to the time periods over which they are evaluated. In addition, A/R removal intensity may diminish when forest expands, especially under increased land competition. This extensive margin response stems from Ricardo's Law of Rent, which states that the most productive land is used first, so that, all else being equal, marginal expansion into lower-productivity land drives down the mean productivity[46]. While purpose-grown energy crops may also have extensive margin responses when production expands, the yield for purpose-grown energy crops is presumed to increase over time, with price-induced yield intensification also contributing[51].

Third, BECCS removal intensity is sensitive to feedstock and technology choices. There is significant potential for BECCS to be produced from feedstocks with minimal direct land use impacts. Across our mitigation scenarios, MSW & residue could supply 53 (37–61) EJ yr⁻¹ of primary energy, with 2.1 (1.5–3.3) GtCO₂yr⁻¹ of the biogenic carbon captured and stored via BECCS. However, our sce-narios show that, over the full study period, 27% (12–39%) of the bio-mass was consumed in sectors in which biomass could not be coupled with CCS. Strategies that encourage increased MSW & residue con-sumption and facilitate a transition to bioenergy used in conjunction with CCS technologies could potentially increase net carbon removal with smaller environmental implications. Generally, a higher land removal intensity for BECCS could be achieved by increasing the uti-lization of waste and residue-based BECCS, adopting CCS technologies earlier, accelerating the transition to primary biomass use in sectors with higher carbon capture potential, and increasing the carbon cap-ture rate of BECCS technologies.

Fourth, A/R removal intensity is sensitive to land management and policy choices. A/R requires expanding forest land cover to augment carbon storage. The efficacy of A/R for carbon removal depends, in part, on the carbon density of the land converted to forest, since only the increase in carbon storage (not the total) is additional. If land-system mitigation policies exclusively prioritize forest expansion (A/R-focused scenarios), excessive depletion of non-forest natural areas could occur, as land use changes are not primarily reflecting differ-ences in carbon storage potential between land cover types. This could potentially result in less efficient land use for carbon removal com-pared to a more comprehensive policy (Main scenarios) that values carbon in all types of land. In addition, our scenarios assume no risk of unplanned reversal of carbon storage on land, but in the real world, the viability and efficiency of A/R depend on how effectively carbon is stored over time, which in turn depends on the quality of monitoring, reporting, and verification (MRV) protocols. Effective land-system mitigation depends on concerted and coordinated efforts among dif-ferent stakeholders to minimize unintended consequences and enhance A/R initiatives in areas with greater potential for land carbon improvement.

Lastly, there could be tradeoffs between removal effectiveness and agricultural price responses. Our scenario analysis shows that land-based CDR measures could lead to higher agricultural prices, consistent with recent studies[22,52]. We found that the magnitude of the agricultural price impact varied with the strength and scope of the policy, primarily the extent to which carbon pricing extended to land, and to a lesser extent how it was differentiated across land types, both of which determined its impact on agricultural land. In general, land pricing transmitted more impact to agricultural markets than energy system pricing alone. Designing land policies to mitigate impacts on agricultural markets, for example, by only subsidizing forest carbon, may be possible but could result in considerably reduced land removal

intensity, as other natural land with relatively higher carbon density may be converted, resulting in carbon leakage and possibly other unintended consequences such as biodiversity loss[53]. Indeed, we find a generally positive relationship between agricultural prices and the land removal intensity of LULUCF (Section S3.4), given that both are sensitive to cropland conversion, suggesting that such trade-offs may be difficult to avoid completely in land system mitigation. Therefore, land-based mitigation policies should be carefully designed, considering not only the trade-offs in carbon removals but also the broader implications for food security, the environment, and overall sustainability.

Our study also provides insights relevant to those developing IAM models and generating new scenarios, and it could help to inform future model intercomparison efforts. Specifically, the implementation of land-system mitigation policies differs significantly across models. Projections from our scenarios exhibited a reasonable range of variation compared to the results from AR6 pathways regarding land-based removals, energy system carbon prices, land removal intensity, and agricultural prices. It is plausible that harmonizing land-based mitigation policies and related assumptions among IAMs could enhance the level of agreement in their projections (see Section S3.4 for discussion of future research). Toward this end, this study establishes a foundation for further assessing the impacts of carbon mitigation measures and for exploring the implications of trade-offs, including environmental consequences[53,54], in the context of broader climate change mitigation.

## Methods
### GCAM description
The Global Change Analysis Model (GCAM)[55] is a widely used open-source global multisectoral economic equilibrium and integrated assessment model (see detailed model documentation at http://jgcri.github.io/gcam-doc/). The model has a detailed market representation of the energy, agriculture, land, and water sectors and their intersectoral connections. GCAM is actively maintained and improved over time. The AR6 scenarios presented in Fig. 1 include $n = 29$ pathways from earlier versions of GCAM, i.e., v4.2 ($n = 1$), v5.2 ($n = 1$), and v5.3 ($n = 27$). This study employs GCAM v6 and incorporates key data and assumption updates to improve the modeling of bioenergy with carbon capture and storage (BECCS) and land-system carbon mitigation policy (Section S2). The model is calibrated to the base year 2015 and runs in 5-year time steps to 2100, driven by future changes in socioeconomic, technological, or policy conditions. The reference scenario uses population and income projections in the SSP2 "Middle-of-the-Road" scenario[56,57]. GCAM aggregates the world into 32 geopolitical regions (Table S1) and uses a logit-based Armington approach[6] to connect and differentiate regional energy and agricultural markets.

The model has a comprehensive depiction of energy flows, from resources (fossil, uranium, or renewables) to energy carriers (electricity, refined liquids, hydrogen, gas, and district heat) and end-use sectors (building, transportation, and industry). Agricultural production and land allocation are modeled at the intersection of 235 water basins and 32 geopolitical regions. The model includes 21 crop sectors (not including dedicated energy crops), 6 livestock sectors, and a managed forestry sector (Table S2), which are aggregated representations of all agricultural commodities included in the FAOSTAT database[58]. In addition, two generic purpose-grown energy crops, woody and herbaceous, are introduced in 2025. Future agricultural productivity is jointly determined by (1) exogenous drivers that imply total factor productivity growth and (2) endogenous productivity responses that are realized via production technology transformations, i.e., intensifications through the use of irrigation, more fertilizer, and more intensified livestock systems. GCAM includes all land covers and employs a nested logit approach to model their competition (Section S2.2). The greenhouse gas (GHG) emissions, including $CO_2$,

$CH_4$, $N_2O$, and F-gases, are traced endogenously as their emission factors are linked to activities in the energy, agriculture, and land systems. The GCAM energy system includes BECCS technologies in refining, electricity generation, hydrogen production, and industry as technology options for carbon dioxide removal (CDR). The model also includes direct air capture with carbon storage (DACCS), which is not included in this study. We break down the Energy and Industrial Processes (EIP) emissions into Fossil Fuels and Industry (FFI) and BECCS for carbon emissions accounting and reporting. The total carbon emissions are then calculated as the sum of FFI, BECCS, and land use, land-use change, and forestry (LULUCF) emissions.

In GCAM, land-system mitigation policies are implemented as a carbon rent to credit landowners for holding carbon stocks[16,30], consistent with the nested land allocation approach[59]. With land-system mitigation policies, landowners receive an annualized land carbon subsidy in addition to rental profits from existing economic activities for managed land or a shadow rental price for natural or unmanaged land. As a result, landowners are incentivized to convert low-carbon-density land to relatively higher-carbon-density land. Land rental profits, derived from production technology specifications and market information, connect land competition and the land mitigation policy to other market-mediated responses. How agricultural and energy markets respond to the land-system mitigation policy plays a crucial role in determining the effectiveness of the policy. A detailed documentation of the land allocation method, land-system mitigation policy, and the related improvements in data and modeling is provided in Section S2.

### Differentiating mitigation efforts by sectors or technologies
In the reference scenario (see Section S3.1 for detailed discussions), in the absence of mitigation measures, the net total cumulative carbon emissions in 2020–2100 are 4380 $GtCO_2$, including 111 $GtCO_2$ from LULUCF, and the temperature rise could reach 3.5 °C by the end of the century[60]. In mitigation scenarios, a trajectory of global carbon prices is implemented to induce changes in the behavior of producers and consumers, collectively aiming to achieve a climate target. Instead of directly targeting climate variables, our study employs a cumulative carbon emissions target-finder to circumvent uncertainties associated with translating emissions and other forcers to climate outcomes[38,61], aligning with recent studies[23]. Following the IPCC AR6 report, we set cumulative carbon emissions targets of 1150 $GtCO_2$ and 500 $GtCO_2$ for 2020–2100 to represent 2 °C and 1.5 °C pathways, respectively. In GCAM v6, the global carbon pricing starts in 2025 with a Hotelling rate of 3% per year, and the model employs an exponential phase-in during the first two model periods. The model finds the optimal 2025 carbon prices and Hotelling paths that reduce reference carbon emissions to achieve the cumulative emission targets. Note that in our study, non-$CO_2$ GHGs are not directly priced. However, GCAM addresses these emissions by utilizing marginal abatement cost (MAC) curves for major non-$CO_2$ GHG species[62]. Specifically, the emission intensity of a given non-$CO_2$ GHG declines with higher GHG prices, which are linked to carbon prices through Global Warming Potential (GWP) values using a 100-year time horizon (GWP-100).

The main scenarios (2 °C Main) differentiate the mitigation efforts between land and EIP sectors to explore how 2 °C futures are achieved with different strengths of land-system carbon pricing. Specifically, land carbon prices, $\beta_t^{Land}$, are linked to EIP carbon prices ($\beta_t^{EIP}$) through a strength parameter ($\mu$) where $\beta_t^{Land} = \mu \beta_t^{EIP}$. We test values of $\mu$ equal 0, 10%, 50%, and 100%. Another set of scenarios (A/R-Focused) builds upon the 2 °C Main scenarios but further differentiates mitigation efforts based on land type. Instead of valuing carbon storage on all land, these scenarios incentivize carbon storage only on forest land, i.e., $\beta_t^{Land:Forest} = \mu \beta_t^{EIP}$ and $\beta_t^{Land:NonForest} = 0$. In addition, it is important to note that primary lignocellulosic bioenergy (or BECCS) is often constrained in IAM scenarios due to sustainability or macroeconomic

concerns[63]. GCAM also has a default strategy that limits the percentage of GDP that can be used to credit BECCS for carbon mitigation (Section S2.8). To achieve this, a carbon pricing penalty ($\beta limit_t^{BECCS}$) is applied, lowering the shadow prices of carbon applied to BECCS in any sector, i.e., $\beta_t^{BECCS} = \beta_t^{EIP} - \beta limit_t^{BECCS}$. The third set of scenarios (Low-Bioenergy), also building upon the 2 °C Main scenarios, imposes a more stringent constraint on primary lignocellulosic biomass, limiting it to 100 EJ in 2100 (increasing linearly prior to 2100), thereby further differentiating the incentives applied to BECCS versus other technologies. In all scenarios, the model solves for EIP carbon prices ($\beta_t^{EIP}$), with other carbon prices linked as discussed above. This is also the case in the last set of scenarios, where the model finds higher $\beta_t^{EIP}$ compared to the Main 2 °C scenarios to achieve the more stringent climate target of 1.5 °C. The solved carbon prices are shown in Fig. 2.

## Deriving and decomposing land removal intensity

The land removal intensity, also referred to as "mitigation density" in Roe et al.[4], is calculated by dividing the cumulative carbon removal by the corresponding land requirement of a land-based mitigation measure. The metric has been widely utilized, particularly in sectoral studies, to compare land-based natural climate solutions[20,64]. However, terminal land use, i.e., land use change in the last period compared to the initial period, is usually used as the land requirement, which may overlook the changes in land use trajectory.

In our study, to ensure a consistent comparison between land-based mitigation measures, e.g., BECCS and afforestation/reforestation (A/R), we define land removal intensity ($RI_T$) over a specific period, $T$, as a ratio between the cumulative carbon removal ($\sum_{t=1}^{T} CDR_t$) and the cumulative land use change ($\sum_{t=1}^{T} LUC_t$), i.e.,

$$RI_T = \sum_{t=1}^{T} CDR_t / \sum_{t=1}^{T} LUC_t \tag{1}$$

Here, $CDR_t$ represents the carbon removal in year $t$ and $LUC_t$ refers to the land use change relative to the initial study year, 2020. The cumulative land use change implies the total net land used for carbon removal in the study period, measured in *hectaer · year* ($ha \cdot yr$). The cumulative removal is measured in metric tons of $CO_2$ ($tCO_2$). Thus, the removal intensity is expressed in $tCO_2\ ha^{-1} yr^{-1}$. The metric can be interpreted as the ratio of interannual mean values, where the mean removal ($tCO_2\ yr^{-1}$) is divided by the mean land requirement ($ha$), i.e., $RI_T = \left( \frac{\sum_{t=1}^{T} CDR_t}{T} \right) / \left( \frac{\sum_{t=1}^{T} LUC_t}{T} \right)$. The metric provides a measure of the effective carbon removal a hectare of land used for CDR can deliver per year, on average, during the study period.

Compared to the conventional approach that uses terminal land use change ($LUC_T$) in the denominator, the approach adopted here factors in the temporal patterns in land use. This is particularly important when comparing BECCS and A/R in integrated assessments, as they exhibit different temporal patterns in land use and carbon removal (see Section S3.3 for detailed discussions).

In our study, the aggregate land removal intensity (e.g., Fig. 3E) is calculated considering both BECCS and LULUCF in $CDR_t$ ($CDR_t = BECCS_t + LULUCF_t$) and including combined areas of energy cropland and forest in $LUC_t$ ($LUC_t = LUC_t^{energy\ cropland} + LUC_t^{forest}$). The aggregate land removal intensity can be decomposed by $RI_T^{BECCS}$ (i.e., $\sum_{t=1}^{T} BECCS_t / \sum_{t=1}^{T} LUC_t^{energy\ cropland}$) and $RI_T^{LULUCF}$ (i.e., $\sum_{t=1}^{T} LULUCF_t / \sum_{t=1}^{T} LUC_t^{forest}$). The metric aims to link removals to their land implications. However, recognizing that not all BECCS are equally land-intensive, and not all energy crops are used in combination with CCS, we further decompose the BECCS removal intensity by feedstock source and land use, distinguishing between

waste & residue-based BECCS (no direct land implications), energy cropland for bioenergy used in sectors in which it could not be coupled to CCS, and energy crop-based BECCS vs. energy cropland for bioenergy used in sectors in which it could be coupled to CCS (e.g., Figs. 3F and S26). Note GCAM traces primary bioenergy demand by sector (such as electricity, refining, gas, hydrogen, and end-uses) and by CCS technology deployment (Fig. S19). However, demand sectors, which vary in their use of BECCS, are not able to distinguish biomass by supply sources (e.g., MSW, residues, or purpose-grown). In order to attribute the supply sources to demand sectors, biomass supply shares by source (Fig. S18) are applied consistently to each demand sector/CCS combination in every region and year (Fig. S20). In addition, when energy cropland is not distinguished (by whether the biomass produced is used in conjunction with CCS), changes in removal intensity take into account the changing share of bioenergy used in conjunction with CCS (i.e., sectoral transition effects).

Since the same primary biomass feedstock mix is applied across demand sectors in GCAM, this same approach is also applied to AR6 pathways for separating waste & residue-based BECCS. However, we cannot further decompose energy cropland into the share used for BECCS versus bioenergy without CCS for AR6 pathways due to limited data available. Furthermore, the land removal intensity can be more uncertain in scenarios involving net deforestation. As our default land mitigation policy covers all land, e.g., including non-forest natural land protection and restoration, we also explore an alternative approach that attributes LULUCF to both forest and other natural lands (Section S3.3). However, this alternative approach is not compared with the AR6 projections due to the relatively lower reporting quality of non-forest natural land and the potential inconsistencies in its definition.

## Isolating effects of mitigation policy on crop prices

To analyze the transmission of prices from carbon markets to agricultural sectors, we build regression models to explore the relationship between changes in staple crop prices and the underlying policy drivers (Eq. 2). The dependent variable is the logarithmic price changes of staple crops ($Y_{p,t}$) relative to the reference scenario (i.e., reference = 1) across mitigation pathways ($p$) and model periods ($t$). The explanatory variables consist of the policy drivers, specifically the various shadow prices of carbon discussed above. We did not include intercept or pathway fixed effects as the carbon prices are the sole differentiating factor between the mitigation pathways and the reference scenario. The error terms are denoted as $\varepsilon_{p,t}$. Consequently, the dataset has 240 observations, i.e., 15 unique mitigation pathways by 16 model periods (see Figs. 6A and S13).

$$\log\left(Y_{p,t}\right) = b^{FFI}\beta_{p,t}^{FFI} + b^{BECCS}\beta_{p,t}^{BECCS} + b^{Land:Forest}\beta_{p,t}^{Land:Forest} + b^{Land:NonForest}\beta_{p,t}^{Land:NonForest} + \varepsilon_{p,t} \tag{2}$$

The results of the regression models, including Eq. 2 (referred to as Model 1) and alternative model specifications, are presented in Table S7. In Eq. 3 (corresponding to Model 4 in Table S7), a slope dummy variable, $S^{A/R-Focused}$, is introduced. And $S^{A/R-Focused} = 1$ when A/R-Focused scenarios are implemented and $S^{A/R-Focused} = 0$ when all land carbon is included in land mitigation policies. Note that Eq. 2 and Eq. 3 are equivalent in terms of explaining variations in $Y_{p,t}$. However, results from Eq. 3 are shown in Fig. 6C as $b^{EIP}$ may be more consistent with a broader literature. Furthermore, as shown in Models 6–8 in Table S7, models with missing policy drivers were also tested.

$$\log(Y_{p,t}) = b^{EIP}\beta_{p,t}^{EIP} + blimit^{BECCS}\beta limit_{p,t}^{BECCS} + b^{Land}\beta_{p,t}^{Land} + b^{Land'}S^{A/R-Focused}\beta_{p,t}^{Land} + \varepsilon_{p,t} \tag{3}$$

Compared to the decomposition approach developed by Stehfest et al.[65] and applied in Fujimori et al.[22], our approach to partitioning price impacts relative to a reference scenario is more internally consistent.

## Data availability

The data from AR6 Scenario Database used in this study are available at data.ene.iiasa.ac.at/ar6/. The GCAM simulation results and processed data generated in this study are available at zenodo.org/record/8244015 (https://doi.org/10.5281/zenodo.8244015).

## Code availability

GCAM is an open-source model at github.com/JGCRI/gcam-core, and the specific version of the model used in this study is archived at github.com/realxinzhao/paper-nc2024-LandBasedCDR-GCAM (https://doi.org/10.5281/zenodo.10659353). The R code for generating main figures is available at github.com/realxinzhao/paper-nc2024-Land-BasedCDR-DisplayItems (https://doi.org/10.5281/zenodo.10659392)[66].

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

## Acknowledgements

The PNNL authors acknowledge support from the ExxonMobil Technology and Engineering Company. The views and opinions expressed are those of the authors alone. We appreciate Matthew Binsted for managing the project in its later stages. We extend our gratitude to Kate Calvin, Jae Edmonds, Ben Bond-Lamberty, Yang Ou, Jay Fuhrman, Pralit Patel, Dalei Hao, Jon Sampedro, Patrick O'Rourke, Brinda Yarlagadda, Xueyuan Gao, Alan Di Vittorio, Kanishka Narayan, Ryna Cui, and Page Kyle for their valuable comments and suggestions.

## Author contributions

X.Z., B.K.M., M.A.W., and H.C.M. conceptualized the research. X.Z. led the modeling and simulations and wrote the first draft of the manuscript. X.Z. and B.K.M. led the analysis. X.Z., B.K.M., M.A.W., and H.C.M. contributed to the interpretation of the results and writing the manuscript.

## Competing interests

The authors declare no competing interests.
