## [Peer Review File · Nature Communications]

Trade-Offs in Land-Based Carbon Removal Measures under 1.5°C and 2°C FuturesREVIEWER COMMENTS

Reviewer #1 (Remarks to the Author):

GENERAL COMMENTS

In this paper, the authors describe their research using the Global Change Analysis Model (GCAM) to study trade-offs between two forms of land-based carbon dioxide removal (CDR): afforestation/reforestation (A/R) and bioenergy with carbon capture and storage (BECCS). This is an important topic, as land-based CDR figures prominently in most pathways to meet global climate targets and is often projected to involve vast amounts of land, with significant impacts on food prices, biodiversity, etc. Figuring out how different policies would affect total land-based CDR and its various side effects/co-benefits is a valuable contribution to our understanding of climate policy.

I recommend asking the authors to revise & resubmit the paper. There are one or two main issues and a handful of small ones. I am confident the authors can address all of them.

While the research underlying the paper appears solid, I find the presentation of and discussion of its findings both need improvement. I think the presentation tends to be cluttered with details and comparisons to the literature that make it hard to follow the main story. It takes considerably more work than it should, on the reader's part, to figure out paper's substantive take-home points, and the Discussion could more cleanly separate information most relevant to other researchers from information most relevant to policymakers. Some parts of the discussion (e.g., lines 315-318) are admirably clear in this respect, but there is still room for improvement. I also think that it would help if the authors offered more synthesis of their findings here, rather than focusing on broad statements or narrow analysis. See below for examples of passages that detract from the clarity of the paper.

The figures are detailed, informative, and well-executed.

MAJOR ISSUES

1. As mentioned in my general comments, the overall presentation of the research findings and their significance could be much clearer. After reading the paper and reviewing key sections and the SI, it remains unclear what the big take-home lessons are, other than the very general point that there are trade-offs in land-based CDR methods and that these can affect agricultural prices. A few illustrative examples of places that strike me as cluttered or unclear:

* In 98-99: while it is important to know that the current study's projections for LULUCF and BECCS are consistent with the literature (In. 98-99), that is obviously not the main thing to know about trade-offs between land-based CDR options.

* In 133ff: this section is supposed to be about the impact of land carbon pricing strength on CDR in 2C futures, which is one of the central topics of the paper. Yet, the first twenty lines of the section gives a detailed summary of land-based CDR in just one scenario, rather than leading with a comparison or analysis of the impact that the different land carbon pricing policies have on CDR.

* In 253ff: this section discusses the impact of land-based CDR on agricultural prices, which is another important topic for the paper. The first paragraph, though, gives us the broad range of price changes and compares them to the literature, rather than leading with the important and methodologically interesting findings about the impact of mitigation measures on crop prices and the decomposition of that impact across sectors.

These might seem like nitpicky examples or a stylistic complaint, not a major issue, but the cumulative effect of choices like these is to make the paper much harder to follow and understand than it needs to be.

2. On a possibly more substantive issue, I'm confused about how the authors calculate the land carbon removal efficiency for BECCS. Specifically, I can't tell whether they are using the total land area used for purpose-grown bioenergy crops or just the land area used for purpose-grown bioenergy crops that are then used for BECCS. The Methods section seems not entirely clear about this, but some statements in the paper (e.g., lines 227-234, where increasing deployment of CCS is said to affect land carbon removal efficiency) suggest it is the former, which I find puzzling. While such a method makes sense for calculating the "mitigation density" of land, as in Roe et al. (2021), it doesn't make sense to me for calculating land carbon removal efficiency. Suppose that A/R carbon removal were zero and that waste/residue-based BECCS were ignored. If I want to know how many hectares of land it takes to remove a ton of CO₂, it seems to me I wouldn't want to include the hectares of land used to grow biofuels that were not used with CCS. Right? For example, it seems to me that increasing the amount of land used to grow corn for ethanol manufactured without CCS (without changing the quantity of biofuels used with CCS), for instance, shouldn't affect the land carbon removal efficiency of the BECCS industry. Since land carbon removal efficiency plays a significant role in the paper, I hope the authors can clarify their method of calculation and, if necessary, either (1) justify including biofuels without CCS in the calculation or (2) revise their calculations.

MINOR ISSUES

* In 38-39: when discussing removals from LULUCF and BECCS, it's worth noting here that the reason these approaches dominate CDR in the AR6 scenarios is that very few IAMs had any other form of CDR at the time

* In 103: this might be the most nitpicky comment I've ever written in a review (ha! just kidding!), but does the No-LCP scenario really show a "trade-off" between LULUCF and BECCS, or just an inverse relationship? I take it the mechanism for the inverse relationship is that in the absence of a C price, removals via LULUCF decline and then reverse, requiring a higher carbon price to stay within the prescribed carbon budget, which incentivizes more BECCS. Trade-off seems the wrong word for this in the No-LCP scenario, as 'trade-off' suggests to me that increasing one causes a reduction in the other.

* In 203: can you explain in more detail why energy cropland removal efficiency decreases in the Low-Bioenergy scenario? Is it because more purpose-grown bioenergy crops are used without CCS when biofuels become more expensive?

* Fig. S19: I find the use of negative values for bioenergy without CCS counterintuitive and unhelpful. The legend for CCS/No CCS is also unclear. This isn't a big deal, obviously, but it might be clearer if the No CCS values were included as positive values but were hatched or outlined instead of fully shaded.

Reviewer #2 (Remarks to the Author):

This paper provides noteworthy results by homogenizing mitigation policies for reforestation/afforestation and BECCS strategies, and comparing effects on land use, LULUCF, food prices... Minor revisions are suggested below.

General comments:

The paper would gain in providing a more detailed definition of the baseline. It seems that any land would be available for CDR, is that really the case? Is the baseline of your model considering the ongoing and projected deforestation, and forest/biodiversity protection? What would be the

counterfactual of limited action, for instance, creating natural preserves, and then let the nature take over? What could be the results in term a carbon sequestration and biodiversity?

The definition of BECCS would gain to be more detailed. The direct comparison of BECCS vs A/R suggests that benefits are 1 to 1, it might be true for carbon storage, but co-benefits might make the reality more complex. How does BECCS and energy crops compare with the higher biodiversity of the forest and associated benefits (resilience, resistance to diseases...)? Due to the difficulty to attribute costs to some co-benefits, is there a way to qualitatively include co-benefits and trade-offs into discussions throughout the document?

Has the entire system for BECCS been assessed? Where would the CO₂ be stored, and what is the cost and carbon intensity of that storage? And would the CO₂ and/or the biomass have to be transported, and what is the cost and carbon intensity of that transportation? Under which condition does a BECCS system actually result in negative carbon emissions? Also, more recent studies also talk about BiCRS, how would BiCRS (and the variety of products it can create) differ from BECCS in your study?

Detailed comments:

Abstract: Define / spell out GCAM and IAM

Figure 1: the color choice makes it difficult to distinguish the 2C main scenario and the 2C A/R focused scenario.

Line 52: the authors point out the statistically negative relationship between the cumulative removals from BECCS and LULUCF. Are these two CDR methods also competing with land for food production? Or other types of land use (like pristine land)?

Line 72-76: this sentence seems counter intuitive and might require additional explanations. Does it mean that maintaining and preserving our forests would not be as efficient than having energy crops for BECCS, from a carbon capture and storage viewpoint?

Line 78: how do monitoring, reporting, and verification protocols vary? How do monitoring, reporting, and verification differ between various CDR methods?

Line 85: could you give a couple of examples to understand better what is behind "land" and "energy carbon". Does land refer to A/R, and energy carbon refer to crops for BECCS?

Line 90: could you explain in greater details how the land carbon pricing works? What is the cost of A/R and of BECCS? What is the value given to the land carbon pricing? Would the land carbon pricing fully pay for the deployment of these technologies? Could it be a source of revenue for land owners? Would a land carbon pricing that is too high compete with food production?

Line 112: this paragraph compares A/R-focused scenarios with the main scenario that has relatively more BECCS. Does this study account for potential displacement of fossil fuels by higher energy production with BECCS (and potentially transforming current fossil fuel power plants in BECCS facilities)? Does it also account for the increasing energy demand?

Line 121: is FFI carbon prices similar to the 45Q tax credit (tax credit in the United States recently increased by the Inflation Reduction Act)?

Line 137: Do the forestry residue come from A/R? Are the A/R scenario combined with the BECCS scenario with for instance forest thinning for fire mitigation?

Line 151: if A/R and BECCS lead to higher agricultural prices, is there a risk of food shortage? The need for food will necessarily conflict with climate goals. If there is a risk of food shortage, how realistic are these scenarios? Also, do these estimates consider a growing global population?

Line 156: "In the reference scenario, global non-energy cropland expands over time", this means that non-energy cropland is taking over some other land uses, which type of lands are these? Land for food production? Forests? Other natural habitats?

Line 158: "Global net deforestation switches to net afforestation". The "net" metric might need to be discussed, as afforestation is difficult to achieve with success. Numerous afforestation projects have seen low success rate with a high mortality of trees, and little oversight, and not necessarily prioritizing native species. Please provide more details on how reforestation and afforestation plan to be deployed.

Line 192: what makes pasture available for conversion? Is that a type of land projected to be less needed for livestock in the future?

Line 269-270: this is a major conclusion, encouraging efforts from all sectors and the deployment of a portfolio of solution as also suggested by your statement lines 278-280.

Line 329: could natural land with relatively high carbon density be protected, to improve ecosystems conservation and reduce the risk of conversion?

Line 376-377: this sentence might have to be revised. "derived" or "based on" should be removed.

Reviewer #3 (Remarks to the Author):

Please see the document attached.

Referee Report “Trade-Offs in Land-Based Carbon Removal Measures under 1.5°C and 2°C Futures”

The paper presents a comprehensive study using the Global Change Analysis Model (GCAM) to investigate the trade-offs between two major land-based carbon dioxide removal (CDR) strategies: afforestation/reforestation (A/R) and bioenergy with carbon capture and storage (BECCS). The study adeptly utilizes various policy scenarios to explore these trade-offs, focusing on aspects such as land competition, carbon pricing, and impacts on agricultural prices. The findings contribute to a deeper understanding of how land-based CDR strategies might perform under different global warming targets and policy frameworks.

Key Findings:

Trade-offs between A/R and BECCS: The study identifies a trade-off between A/R and BECCS, as both are land-intensive and compete for the same resource. An inverse relationship exists between their deployment, with a partial valuation of land carbon storage enabling a net land sink even under large-scale BECCS deployment.

Impact of Carbon Pricing on Land Use and CDR: Various carbon pricing scenarios show that higher land carbon prices lead to increased land carbon storage and reduced land use for food and energy. Conversely, weaker land mitigation policies result in the opposite effect. This variation significantly affects the land-based CDR strategies deployed.

Effects on Agricultural Prices: The scenarios indicate that land-based CDR measures, particularly when associated with strong land carbon pricing, could lead to significant increases in agricultural prices. Demand-side adaptations and yield intensifications could mitigate this increase. There is a complex interaction between land-system mitigation policies, agricultural markets, and food security.

Policy-Driven Variations in Land Use and CDR Efficiency: The study explores different policy scenarios, including those that focus exclusively on forest carbon or impose constraints on bioenergy. These scenarios demonstrate how policy choices can significantly affect land use patterns and the efficiency of land-based CDR measures.

Land Carbon Removal Efficiency: The paper calculates land carbon removal efficiency, considering both BECCS and A/R, and finds significant variations across scenarios. The efficiency of BECCS improves over time due to yield growth and technological progress, whereas the efficiency of A/R tends to decline as forests mature.

Strengths:

- The paper effectively utilizes GCAM to explore complex interactions between land-based CDR measures and policy scenarios.
- It comprehensively analyzes the trade-offs between A/R and BECCS, highlighting critical aspects like land competition, carbon pricing, and agricultural prices.
- The study contributes valuable insights into policy-driven variations in land use efficiency and their implications for global carbon budgets and agricultural markets.

Weaknesses:

- While the study accounts for various policy scenarios, it might not fully capture the complexities of real-world policy implementation and socio-economic dynamics.
- The paper focuses on the efficiency of land-based CDR methods but might not sufficiently address these strategies' potential environmental and social impacts.
- Though robust, reliance on a single modelling framework might limit the generalizability of the findings, as different models might yield different outcomes under similar scenarios.

Comparison with Existing Literature: The study aligns with the current discourse in the field, particularly regarding the trade-offs and interactions between A/R and BECCS strategies. Its focus on policy-driven variations in land use and CDR efficiency adds valuable insights to existing literature, especially regarding the potential impacts on agricultural prices and land use patterns. However, while the study extends the discussion on land-based CDR strategies, it does not significantly diverge from existing approaches or introduce novel methodologies that would distinguish it substantially from previous research. The manuscript's findings, while insightful, align closely with established understandings in the field, as outlined in the Intergovernmental Panel on Climate Change (IPCC) reports and other comprehensive studies.

Recommendations:

Broader Contextualization: To enhance the manuscript's appeal, it could benefit from a broader contextualization of its findings within the global climate change mitigation landscape. This might include comparing these land-based strategies with other mitigation options or considering policy choices' socio-economic and environmental implications more comprehensively.

Comparing methodologies: This is a big ask, but incorporating or comparing results from different models or methodologies could provide a more robust and diversified analysis, strengthening the manuscript's contribution to the field. By comparing and contrasting results from these models with those from GCAM, researchers can gain insights into the robustness of their findings and explore a broader range of scenarios and policy implications. Each model has its own strengths and limitations, and by using them in conjunction, it's possible to provide a more nuanced understanding of the complex interactions between human activities, energy systems, and the environment. For instance, while GCAM provides a detailed representation of the energy and land-use sectors, MESSAGE or WITCH might offer different perspectives on technological change and innovation in the energy sector. Similarly, comparing the socio-economic implications of climate policies using both GCAM and REMIND or AIM/CGE could yield a more comprehensive view of the potential impacts on different regions and economic sectors.

Suitability for Nature Communications:

While the research is methodologically sound and the findings are indeed valuable, the scope of the study, focusing primarily on the trade-offs between two specific land-based CDR strategies under a limited set of policy scenarios, may be too narrow for a broad-based, multidisciplinary journal like Nature Communications. Nature Communications often seeks research that has wide-ranging implications, novel approaches, or ground-breaking findings that can appeal to a broad scientific audience. In this context, while rigorous and well-executed, the manuscript might not meet the criteria for a substantial contribution to the field in a way that would justify publication in such a high-impact journal.

Given the manuscript's current scope and focus, it might find a more suitable audience in a journal specializing in environmental science, climate policy, or land use studies. Such a platform could provide a more targeted audience that would appreciate the specific nuances and contributions of the research.

#	Reviewer #1 Remarks (1 st Round)	Responses/Actions (1 st Round)
A0	General Comments. In this paper, the authors describe their research using the Global Change Analysis Model (GCAM) to study trade-offs between two forms of land-based carbon dioxide removal (CDR): afforestation/reforestation (A/R) and bioenergy with carbon capture and storage (BECCS). This is an important topic, as land-based CDR figures prominently in most pathways to meet global climate targets and is often projected to involve vast amounts of land, with significant impacts on food prices, biodiversity, etc. Figuring out how different policies would affect total land-based CDR and its various side effects/co-benefits is a valuable contribution to our understanding of climate policy. I recommend asking the authors to revise & resubmit the paper. There are one or two main issues and a handful of small ones. I am confident the authors can address all of them.	Thank you for the careful reading of our paper and the valuable comments. All the comments are very clear, spot-on, and insightful! We believe the paper has been significantly improved as a result of the very useful comments and suggestions. Please note that in the response/revision:  • Some comments are reordered to improve the structure & communication of the responses, but all points are included in this table. • Comments/responses are indexed in case of future communications, e.g., A0. • Changes made are highlighted in both the Main (track-change) manuscript and SI (redline). • SI figures are reindexed since new figures were added. Detailed point-by-point responses are provided below.
A1	The figures are detailed, informative, and well-executed.	Thanks.
A2	While the research underlying the paper appears solid, I find the presentation of and discussion of its findings both need improvement. I think the presentation tends to be cluttered with details and comparisons to the literature that make it hard to follow the main story. It takes considerably more work than it should, on the reader's part, to figure out paper's substantive take-home points, and the Discussion could more cleanly separate information most relevant to other researchers from information most relevant to policymakers. Some parts of the discussion (e.g., lines 315-318) are admirably clear in this respect, but there is still room for improvement. I also think that it would help if the authors offered more synthesis of their findings here, rather than focusing on broad statements or narrow analysis. See below for examples of passages that detract from the clarity of the paper.	The points have been well-received. We conducted a full round of updates (making use of the additional space allowed in Nature Comm.) to address the motivations behind scenario design, enhance the paper's structure, and expand the discussion section in line with the suggestions. For instance, key points such as the following were summarized and discussed in the Discussion section.  • BECCS and A/R both contribute significantly to mitigation in 1.5C and 2C scenarios. • The removal intensity of BECCS is typically higher than A/R over long time horizons. • BECCS removal intensity is sensitive to feedstock and technology choices. • A/R removal intensity is sensitive to land management and policy choices. • There could be tradeoffs between removal effectiveness and agricultural price responses. With these, the clarity of communication has been significantly enhanced.
A3	Major Issue 1. As mentioned in my general comments, the overall presentation of the research findings and their significance could be much clearer. After reading the paper and reviewing key sections and the SI, it remains unclear what the big take-home lessons are, other than the very general point that there are trade-offs in land-based CDR methods and that these can affect agricultural prices. A few illustrative examples of places that strike me as cluttered or unclear:	Thanks for the suggestions and detailed examples. Responses to illustrative examples are provided below. As mentioned above, major revisions have been made in the Introduction and Discussion sections to address the comments.
	* In 98-99: while it is important to know that the current study's projections for LULUCF and BECCS are consistent with the literature (ln. 98-99), that is	Thanks. Instead of just showing that the results are consistent with the literature, we also intended to communicate that results from our scenarios were able to

#	Reviewer #1 Remarks (1 st Round)	Responses/Actions (1 st Round)
	obviously not the main thing to know about trade-offs between land-based CDR options.	cover most of the AR6 points in Fig. 1A so that they represent the literature variations in LULUCF and BECCS well. We have made this clear now. The sentence is updated: “Outcomes from our mitigation scenarios with regard to LULUCF and BECCS removals encompass a large share of the variability observed in AR6 pathways (Fig. 1A).” In addition, we also added discussions in comparison to AR6 pathways in the Discussion section. “Projections from our scenarios exhibited a reasonable range of variation, compared to the results from AR6 pathways, regarding land-based removals, energy system carbon prices, land removal density, and agricultural prices. It is plausible that harmonizing land-based mitigation policies and related assumptions among IAMs could enhance the level of agreement in their projections (see Section S3.4 for discussions of future research).” We believe the discussions are useful and important since they indicate our scenario design is meaningful, and the results demonstrate high sensitivity but also are communicable to the literature.
	* In 133ff: this section is supposed to be about the impact of land carbon pricing strength on CDR in 2C futures, which is one of the central topics of the paper. Yet, the first twenty lines of the section gives a detailed summary of land-based CDR in just one scenario, rather than leading with a comparison or analysis of the impact that the different land carbon pricing policies have on CDR.	Thanks! Good point! We believe it is useful and important to provide a relatively more detailed description of the 2C Main scenario (100%-LCP) since it serves as the primary scenario and provides the basis for comparisons both across LCPs and alternative policies. To enhance clarity, we have elevated this portion of the description into an independent section. We have now separated a new section (Connecting land use and carbon removal) from the original section (Impact of land carbon pricing strength on CDR in 2C futures). Note that the original section is also renamed to “Impact of land carbon pricing strength”. This new section provides a more detailed exploration of one scenario (2C Main 100% LCP), focusing on illustrating the connection between land and carbon removal. This also provides an opportunity to explain the decomposition of the land CDR efficiency (renamed to land removal intensity now) in that context. Meanwhile, the original section will place greater emphasis on the broader scenario comparison. Please refer to the updated manuscript for these changes.
	* In 253ff: this section discusses the impact of land-based CDR on agricultural prices, which is another important topic for the paper. The first paragraph, though, gives us the broad range of price changes and compares them to the literature, rather than leading with the important and methodologically interesting findings about the impact of mitigation measures on crop prices and the decomposition of that impact across	We agree with the reviewer that the comparison with the AR6 results should not be the main focus of this section (Agricultural price implications of land-based mitigation measures). This was indeed our intention as more space was used for describing the price impact decompositions. Please note that only one sentence in this section describes the comparison, and we have now improved figure panel

#	Reviewer #1 Remarks (1 st Round)	Responses/Actions (1 st Round)
	sectors.	referencing: “Scenarios with stronger land carbon policing, such as 50%-LCP and 100%-LCP, align more closely with AR6 pathways, although agricultural prices are among the least reported variables in those pathways (Fig. 6B)”
	These might seem like nitpicky examples or a stylistic complaint, not a major issue, but the cumulative effect of choices like these is to make the paper much harder to follow and understand than it needs to be.	These comments are very helpful! The structure and communication of our study have been improved significantly as a result of the comments.
A4	Minor Issue * Fig. S19: I find the use of negative values for bioenergy without CCS counterintuitive and unhelpful. The legend for CCS/No CCS is also unclear. This isn't a big deal, obviously, but it might be clearer if the No CCS values were included as positive values but were hatched or outlined instead of fully shaded.	Thanks. This issue (moved up in order) is indeed important and it is relevant to Major Issue 2 (A5) raised. We updated the color palettes, added outline groups (hatched didn't work well with ggplot, though test code was added to the repo), and improved the annotations (see Fig. RR1 below or the updated Fig. S19). We also added more details in the caption. The code/workflow for generating the figure (and a few alternative versions tested) is also provided in the GitHub repo (realxinzhao/paper-nc2024-LandBasedCDR-DisplayItems).
A5	Major Issue 2. On a possibly more substantive issue, I'm confused about how the authors calculate the land carbon removal efficiency for BECCS. Specifically, I can't tell whether they are using the total land area used for purpose-grown bioenergy crops or just the land area used for purpose-grown bioenergy crops that are then used for BECCS. The Methods section seems not entirely clear about this, but some statements in the paper (e.g., lines 227-234, where increasing deployment of CCS is said to affect land carbon removal efficiency) suggest it is the former, which I find puzzling.	Thanks. The concerns and suggestions are well-received. Yes, the reviewer correctly grasps our approach. We did not differentiate purpose-grown energy land by CCS technologies/sectors (e.g., whether bioenergy technologies were combined with CCS or not). Here are some explanations and clarifications: (1) Our metric was designed to link BECCS on the demand side to land use. The land competition we aim to highlight is between all purpose-grown land and other land types (including forests). Additionally, purpose-grown energy crop production occurs on the supply side, where producers supply to the primary biomass market without knowing how it will be consumed. This is similar to the corn market example, where corn products and prices remain the same regardless of whether they are used as food, feedstuff, or ethanol feedstock. Similarly, CCS deployment on the demand side does not differentiate the biomass market. Thus, the metric aligns with the market equilibrium framework, and our model does not directly provide land results by CCS.(2) Unlike 1G bioenergy (e.g., the corn ethanol example the reviewer used), which usually has nonenergy uses (e.g., food or feed), 2G bioenergy is entirely used for energy. We agree the concern would be bigger and more technical for 1G.(3) When no CCS technology is combined with biomass energy consumption, it can be considered as having 0% carbon capture and sequestration. Therefore, our approach is more general and incorporates impacts from sectoral transitions, such as mitigation policy-induced increases in the share of biomass use in sectors with (higher) CCS, which has been demonstrated to be important subsequently (and it also affects agricultural prices).(4) The current metric can indeed be further decomposed to differentiate purpose-grown energy cropland by CCS technologies/sectors. The land removal

#	Reviewer #1 Remarks (1 st Round)	Responses/Actions (1 st Round)
		efficiency that attributes BECCS to “CCS-designated” purpose-grown energy cropland will be closer to results from sectoral studies. However, further decomposition is subject to technical assumptions to connect purpose-grown energy cropland to CCS technologies, and we cannot compare results with AR6 pathways at that level of detail. After all, we concur with the reviewer's suggestion that presenting more detailed results would be valuable. Accordingly, we now provide additional results to further decompose BECCS land removal intensity (efficiency) by CCS technologies/sectors. We believe they will (1) enhance communication and better illustrate sectoral transitions, (2) strengthen the link between IAM and sectoral analysis, and (3) advance the decomposition and understanding of land removal efficiency. Please refer to the more detailed responses below.
	While such a method makes sense for calculating the “mitigation density” of land, as in Roe et al. (2021), it doesn't make sense to me for calculating land carbon removal efficiency.	Please note that our intention was not to introduce a new metric or naming convention. Our metric (now called land removal intensity) seeks to establish a connection between carbon removals (Fig. 3A, where carbon budget matters) and land use (Fig. 3D, where the land competition matters). However, there are no essential differences between our metric and “mitigation density” in Roe et al. (2021). Please note that the method in Roe et al. (2021) for IAMs also did not differentiate energy cropland by CCS. We believe our method is an improvement (discussed in Methods) due to the following reasons:  (1) We distinguished nonland-based BECCS (2) Rather than using terminal land use, our method uses cumulative land use, which accounts for the nonlinear dynamics of land use change over time. (3) We will be able to differentiate energy cropland by CCS per your suggestions. To avoid confusion, we have now changed the name of the metric from “land carbon removal efficiency” to “land removal intensity” which probably describes the metric better.
	Suppose that A/R carbon removal were zero and that waste/residue-based BECCS were ignored. If I want to know how many hectares of land it takes to remove a ton of CO₂, it seems to me I wouldn't want to include the hectares of land used to grow biofuels that were not used with CCS. Right? For example, it seems to me that increasing the amount of land used to grow corn for ethanol manufactured without CCS (without changing the quantity of biofuels used with CCS), for instance, shouldn't affect the land carbon removal efficiency of the BECCS industry. Since land carbon removal efficiency plays a significant role in the paper, I hope the authors can clarify their method of calculation and, if necessary, either (1) justify including biofuels without CCS in the calculation or (2)	We take action to address this concern by providing more detailed results. But we first add a little clarification on the methods (sectoral vs. IAM analysis) here. As discussed above, the question of “how many hectares of land it takes to remove a ton of CO₂,” this question is more directly pertinent to sectoral analysis, e.g., Hanssen et al. (2020)¹, than to global economic & integrated assessments (our study). In sectoral analysis, the assumption is usually that land and yield are given and dedicated for bioenergy production, and the corresponding demander is known (e.g., electricity with CCS). The calculation would be fairly straightforward as biomass yield (MJ/ha) X carbon density (tCO₂/MJ) X CCS capture rate in a sector (%) = BECCS carbon capture per hectare (tCO₂/ha). This is not the case in our modeling, where the market equilibrium is the focus and many factors are

#	Reviewer #1 Remarks (1 st Round)	Responses/Actions (1 st Round)
	revise their calculations.	endogenous (e.g., crop yield and carbon capture rates). E.g., crop producers, energy sectors (electricity, refining, etc.), or other industries are different agents and operate in different markets. We now try to connect land use to primary biomass demand sectors by CCS in order to provide more detailed results, as the reviewer suggested. As previously mentioned, there was no direct result from the model since consumers cannot distinguish biomass input by supply sources (e.g., MSW, residues, or purpose-grown). To bridge this gap, biomass supply shares (by source) were used to attribute the consumption by CCS to supply sources for each region (trade adjustments were also required), year, and scenario. Detailed results for primary bioenergy by demand sectors, CCS technology deployment, and supply sectors are presented in Fig. RR2 (Fig. S20), which serves as a decomposition of Fig. RR1A by supply sectors. Furthermore, the CCS shares for purpose-grown biomass were used to separate the corresponding cropland by CCS. The decomposition of land-based BECCS removal intensity is refined in Fig. RR3-5 (or updated Figs. 3F, 4F, S26, & S27). In particular, the efficiency of Land-based BECCS is subdivided into BE-NonCCS:Land(NonCCS) and BECCS vs. Land (CCS). BE-NonCCS:Land(NonCCS) has land implications but does not result in net carbon removal (as bioenergy is used in sectors with no CCS deployment). In contrast, BECCS vs. Land (CCS) illustrates the relationship between energy crop-based BECCS and CCS “associated” energy cropland. The results unveil more consistent removal intensity (larger compared to values before the decomposition) for land-based (CCS) BECCS, i.e., 11.1 – 11.8 tCO₂/ha/yr for 2C scenarios and 12.9-13 tCO₂/ha/yr for 1.5C scenarios. Notably, the sectoral transition of primary biomass use from NonCCS sectors to CCS sectors (horizontal lines in the decomposition) is faster when carbon prices are higher (e.g., 1.5C vs. 2C main) as indicated by smaller “NonCCS biomass land”. A number of changes were made in the main text and SI to address the comments and reflect the discussions above. E.g.,  (1) Throughout the result section, more detailed results were added. E.g., the following sentence is added when comparing the Low-Bioenergy scenario with the Main: “As a result, BECCS deployment is smaller, and energy cropland removal becomes less efficient, with the removal intensity decreasing to 7.4 from 8.7 tCO₂ha⁻¹yr⁻¹ (Main), and to 11.1 from 11.8 (Main) when energy cropland that is attributed to biomass used without CCS is removed from the intensity calculation” (2) Related main figures and SI figures are updated. And a new SI figure (S20) was added. (3) Method section update (Deriving and decomposing land removal intensity).
A6	Minor Issue * In 38-39: when discussing removals from LULUCF and	Thanks for the suggestion. The following sentence in the first paragraph (main text)

#	Reviewer #1 Remarks (1 st Round)	Responses/Actions (1 st Round)
	BECCS, it's worth noting here that the reason these approaches dominate CDR in the AR6 scenarios is that very few IAMs had any other form of CDR at the time.	is updated to address this suggestion: “While BECCS was deployed in almost all these pathways, about three-quarters of them also relied on net future LULUCF carbon removals, mainly through A/R, while other CDR methods were not extensively utilized (Section S1).” In addition, we provided the summary statistics of the CDR by sectors/technologies in IPCC AR6 pathways in Fig. S2. We have also added the following clarifications in Section S1, which provides details for AR6 pathways: “While most pathways reported results for BECCS and LULUCF, only a subset of them permitted and reported other CDRs, with 241 pathways including DACCS and 113 pathways involving enhanced weathering. That is, not all models represented all CDR technologies. The reliance on BECCS and A/R removals and energy system mitigation could be less pronounced if higher levels of new CDRs are deployed.”
A7	Minor Issue * In 103: this might be the most nitpicky comment I've ever written in a review (ha! just kidding!), but does the No-LCP scenario really show a “trade-off” between LULUCF and BECCS, or just an inverse relationship? I take it the mechanism for the inverse relationship is that in the absence of a C price, removals via LULUCF decline and then reverse, requiring a higher carbon price to stay within the prescribed carbon budget, which incentivizes more BECCS. Trade-off seems the wrong word for this in the No-LCP scenario, as ‘trade-off’ suggests to me that increasing one causes a reduction in the other.	We provide clarifications around the trade-off questions raised. Original line 103: “This suggests a clear trade-off between LULUCF and BECCS when the strength of land mitigation policy is varied.” We acknowledge that ‘trade-off’ suggests “increasing one causes a reduction in the other”, implying an inverse relationship. In this specific context, increasing (decreasing) LULUCF removal leads to a reduction (increase) in BECCS removal, despite the driver being the strength of land mitigation policy. Our analysis of AR6 pathways revealed a weak negative relationship between BECCS and LULUCF removals (CB was controlled as well), depicted in Fig. 1 and Section S1. The beta coefficient was -0.17 (OLS). Doing the same calculation for using results from our scenarios, the beta coefficient would be -0.86 for all scenarios (with CB controlled) and -2 for 2C Main scenarios. The negative relationship indicates a potential trade-off between LULUCF and BECCS when the strength of land mitigation policy is varied. And our results show a stronger tradeoff, given that the beta coefficients are closer to -1, in contrast to the coefficient implied by AR6 pathways. If the question is more about the role of different carbon prices or carbon budget targets in this comparison, we acknowledge that a fixed carbon budget could potentially contribute to such a trade-off, although not necessarily. However, this aspect does not alter our original descriptions. Indeed, as demonstrated in subsequent sections, the trade-off is mainly driven by land competition rather than carbon budget constraints. We have now added this additional information to the Fig1 caption (or see Fig. RR5 below), updated the original sentence, and added additional descriptions to the same section: “The beta coefficient between LULUCF and BECCS, with CB controlled, stands at -

#	Reviewer #1 Remarks (1 st Round)	Responses/Actions (1 st Round)
		0.17 for AR6 pathways. In contrast, the corresponding value for GCAM scenarios studied is -0.86 (-2 for the 2°C Main scenarios).” Main text (original sentence): “For a given carbon budget, this suggests a clear trade-off between LULUCF and BECCS when the strength of land mitigation policy is varied, driven by resource competition.” Main text (added to the first section in results): “Collectively, our scenarios reveal a more pronounced trade-off between LULUCF and BECCS compared to what is suggested by AR6 pathways, with a beta coefficient of -0.86 (in contrast to -0.17 in AR6). This supports our hypothesis that variations in LULUCF across AR6 pathways are predominantly driven by uncontrolled model and scenario differences rather than land-system mitigation policy choices (Section S1).”
A8	Minor Issue * In 203: can you explain in more detail why energy cropland removal efficiency decreases in the Low-Bioenergy scenario? Is it because more purpose-grown bioenergy crops are used without CCS when biofuels become more expensive?	In this case (100%-LCP), the removal intensity of energy cropland was about 15% lower in the Low-Bioenergy scenario compared to the Main scenario, i.e., 7.4 vs. 8.7 tCO₂/ha/yr. Upon segregating “NonCCS” energy cropland, as suggested in A5, the efficiency difference diminishes (to 5%), i.e., 11.1 vs. 11.8 tCO₂/ha/yr. This was because of the lower share of biomass being utilized in sectors with CCS (66% vs. 74%) since the bioenergy constraint predominantly influenced the latter half of the century, where a more robust sectoral transition towards those employing CCS is expected. The remaining 5% difference can be primarily attributed to the lower energy crop yield in the Low-Bioenergy scenario (199 vs. 215 GJ/ha). Furthermore, primary bioenergy prices were notably lower in the Low-Bioenergy scenario due to decreased demand resulting from specified constraints. The reduced crop prices in the Low-Bioenergy scenario discouraged intensification compared to the Main scenario, leading to a relatively lower crop yield. We have now updated the corresponding description: “The linear constraint has a stronger impact on purpose-grown biomass compared to MSW & residue, as the former is mainly utilized in the latter periods of the century. The mean purpose-grown biomass supply decreases by 66% to 22 EJ yr⁻¹, accompanied by a 64% reduction in energy cropland use. The corresponding BECCS CDR experiences an even larger decrease of 69% (-1.8 GtCO₂ yr⁻¹) due to the slower CCS sectoral transition (e.g., 8 percentage points lower in the CCS sectoral share). The constraint, effectively resulting in weaker incentives for BECCS, dampens the price transmission from the carbon market to the primary biomass market. Although the carbon capture rate in CCS sectors is slightly higher driven by the higher carbon prices, the relatively lower prices of primary biomass discourage crop yield intensification (-7%) as well as CCS sector transition.” In addition, Table RR1 (Table S5 in SI) is also added to provide source data for figures and support the scenario comparisons.

Fig. RR1 (Fig. S19 for both original and new) | Projections of global 2020 – 2100 cumulative (A) and annual (B) primary bioenergy by demand sectors and CCS technology deployment. This figure breaks down the primary bioenergy demand shown in Fig. S17 by the deployment of CCS technology (e.g., NoCCS indicates the energy is consumed in a sector/technology not combined with CCS). In panel (A), the CCS technology deployment rate for bioenergy (i.e., the share of primary bioenergy use that is combined with CCS) is shown at the bottom by sector and the aggregate (weighted mean) values are shown in black at the top. Note that both line type (bar border) and transparency (filled color) are used to distinguish CCS technologies. Bars with dotted lines and darker filled colors indicate demand sectors/technologies with no CCS technologies. See additional details in Section S2.8.

Fig. RR2 (New Fig. S20) | Projections of global 2020 – 2100 cumulative primary bioenergy by demand sectors, CCS technology deployment, and supply sectors. This figure provides a decomposition of Panel A of Fig. S19 by supply sectors. See more detailed captions in Fig. S19.

Fig. RR3 (Updated Fig. 3F) | Impact of land carbon pricing strength on 2 °C futures. Points in panel (E) present the relationship between the 2020 – 2100 mean land-based CDR and the corresponding land utilized so that the slope of the lines is a measure of the 2020 – 2100 mean land removal intensity. The slope of background lines (grey), for reference, is labeled (in blue with a unit of tCO₂ per hectare per year). Panel (F) decomposes panel (E) by land-based CDR sources and the corresponding land use, including waste & residue-based BECCS (vertical lines; no land attribution), energy cropland for bioenergy used in sectors without CCS (horizontal lines; no CCS attribution), energy crop-based BECCS vs. energy cropland for bioenergy used in CCS sectors (slopes annotated in orange), and LULUCF vs. forest (slopes annotated in green). Data source: GCAM simulation results.

Fig. RR4 (Updated Fig. 4F) | Impact of alternative energy and land system policies on mitigated futures under 100% land carbon pricing.

Fig. RR5 (Updated Fig. S26) | Land carbon removal intensity. Points in panel (A) present the relationship between the 2020 – 2100 mean land-based CDR and the corresponding land implication so that the slope of the lines implies 2020 – 2100 mean land carbon removal intensity. The slope of background lines (grey), for reference, is labeled (in blue with a unit of tCO_2 per hectare per year). Panel (B) decomposes panel (A) by land-based CDR sources and the corresponding land use, including waste & residue-based BECCS (vertical lines; no land attribution), energy cropland for bioenergy not used in CCS sectors (horizontal lines; no CCS attribution), energy cropland for bioenergy used in CCS sectors (slopes annotated in orange), and LULUCF vs. forest (slopes annotated in green).

Table RR1 (New Table S5) | Connecting land use to land-based carbon removal. This table summarizes the data and metrics used for connecting land to carbon removal and the calculations of land carbon removal intensity. Values shown are for 2020 – 2100 mean values, if not otherwise stated. Note that the marginal land carbon density is computed as the ratio between land carbon stock change and land use change. In scenarios involving net afforestation, the forest's "marginal land carbon density" should be greater than the land removal intensity (LULUCF vs. forest) in absolute values, indicating additionality.

Variable & Metric		2C Main				A/R-Focused			Low-Bioenergy				1.5C			
		No-LCP	10%-LCP	50%-LCP	100%-LCP	10%-LCP	50%-LCP	100%-LCP	No-LCP	10%-LCP	50%-LCP	100%-LCP	No-LCP	10%-LCP	50%-LCP	100%-LCP
Land-based CDR (GtCO ₂ / yr)	Carbon Budget	14.2				14.2			14.2				6.2			
	BECCS & LULUCF	-3.8	-4.6	-5.8	-6.3	-4.7	-6.3	-7.2	-1.3	-2.9	-4.5	-5.2	-6.7	-7.4	-8.3	-8.7
	BECCS	-6.6	-6.1	-5.0	-4.4	-6.1	-5.1	-4.5	-2.9	-2.6	-2.4	-2.4	-9.3	-8.7	-7.5	-6.7
	BECCS Purpose-grown	-4.4	-4.0	-3.1	-2.6	-4.0	-3.2	-2.8	-1.3	-1.1	-0.9	-0.8	-6.1	-5.5	-4.5	-3.9
	BECCS Residue & MSW	-2.2	-2.1	-1.9	-1.8	-2.1	-1.8	-1.7	-1.5	-1.5	-1.6	-1.5	-3.3	-3.1	-2.9	-2.8
	LULUCF	2.8	1.5	-0.7	-1.9	1.4	-1.2	-2.7	1.6	-0.2	-2.1	-2.9	2.6	1.3	-0.9	-2.0
Primary bioenergy (EJ / yr)	Total	153	146	130	119	147	134	125	67	66	66	66	166	161	146	134
	Purpose-grown	97	90	76	65	92	80	71	29	27	23	22	105	100	86	75
	Purpose-grown CCS	78	71	57	48	72	59	51	23	19	16	14	96	89	75	66
	Purpose-grown NonCCS	19	20	19	17	20	21	20	7	7	8	7	10	10	10	10
	Residue	49	48	47	46	48	47	46	30	32	36	37	53	53	53	52
	Residue Crop	39	39	38	37	38	37	36	21	23	27	28	43	44	43	43
	Residue Forestry	10	10	9	9	10	10	10	9	9	9	9	10	10	9	9
	MSW	8	8	8	8	8	8	8	7	7	7	7	8	8	8	8
Biogenic carbon (GtCO ₂ / yr)	Total	12.9	12.3	11.0	10.0	12.4	11.3	10.5	5.6	5.6	5.6	5.6	14.0	13.5	12.3	11.3
	Purpose-grown	8.2	7.6	6.4	5.5	7.7	6.7	6.0	2.5	2.2	2.0	1.8	8.9	8.4	7.2	6.3
	Purpose-grown CCS	6.5	6.0	4.8	4.0	6.0	5.0	4.3	1.9	1.6	1.3	1.2	8.1	7.5	6.3	5.5
	Purpose-grown NonCCS	1.6	1.6	1.6	1.4	1.7	1.7	1.7	0.6	0.6	0.6	0.6	0.8	0.9	0.9	0.8
	Residue & MSW	4.8	4.7	4.6	4.5	4.7	4.6	4.5	3.1	3.4	3.7	3.8	5.1	5.1	5.1	5.0
	Residue & MSW CCS	3.2	3.1	2.8	2.7	3.0	2.7	2.6	2.1	2.2	2.2	2.2	4.2	4.1	4.0	3.8
	Residue & MSW NonCCS	1.6	1.7	1.8	1.9	1.7	1.8	2.0	1.0	1.2	1.4	1.5	0.9	1.0	1.1	1.2
Land use change w.r.t. 2020 (Mha)	Cropland Energy	476	434	354	302	444	383	344	147	134	117	110	519	479	398	344
	Cropland Energy CCS	381	340	267	223	347	284	247	114	96	79	73	471	429	349	300
	Cropland Energy NonCCS	94	94	87	79	97	99	97	33	37	39	37	48	50	49	44
	Cropland NonEnergy	40	-6	-92	-142	12	-49	-89	131	57	-48	-106	36	-22	-124	-179
	Forest & Other natural land	-247	-130	83	198	-115	163	348	-123	25	209	299	-268	-128	112	232
	Forest	-134	-51	98	181	36	400	648	-38	64	193	256	-150	-51	118	207
	Forest Managed	84	72	52	41	85	101	122	108	88	65	53	81	66	42	29
	Forest Unmanaged	-218	-123	46	140	-50	299	527	-145	-24	128	204	-231	-117	77	178
	Other natural land	-114	-79	-15	16	-150	-237	-300	-85	-39	16	43	-118	-77	-6	25
Land carbon stock change (GtCO ₂ / yr)	Forest	-3.3	-1.6	1.1	2.4	0.0	6.1	9.7	-1.3	1.0	2.9	3.7	-3.0	-1.3	1.3	2.5
	Other natural land	-1.3	-0.8	-0.1	0.3	-1.8	-2.9	-3.6	-0.9	-0.2	0.4	0.6	-1.2	-0.7	0.0	0.3
	Others	1.7	0.9	-0.3	-0.8	0.4	-2.0	-3.4	0.6	-0.5	-1.2	-1.4	1.6	0.7	-0.4	-0.8
(Marginal) Land carbon density (tCO ₂ / ha / yr)	Forest	24.4	30.9	11.2	13.1	-1.3	15.2	15.0	33.8	15.0	15.2	14.3	20.0	25.1	11.0	12.2
	Other natural land	11.2	10.3	5.3	17.4	12.0	12.3	12.1	10.3	4.9	22.7	14.3	10.3	9.7	5.7	12.5
	Others	7.0	6.7	3.6	3.9	3.8	12.2	9.7	4.7	22	5.8	4.7	5.9	5.6	3.4	3.5
Land carbon removal intensity (tCO ₂ / ha / yr)	BECCS & LULUCF	-11.2	-11.9	-12.7	-13.1	-9.8	-8.0	-7.3	-11.8	-14.6	-14.6	-14.3	-18.1	-17.2	-16.2	-15.8
	BECCS (Purpose-grown) & LULUCF	-4.7	-6.4	-8.5	-9.3	-5.4	-5.7	-5.5	2.3	-6.8	-9.5	-10.1	-9.2	-9.9	-10.5	-10.8
	BECCS	-14.0	-14.0	-14.2	-14.6	-13.7	-13.3	-13.0	-19.5	-19.8	-20.8	-21.4	-18.0	-18.1	-18.7	-19.5
	BECCS Purpose-grown	-9.3	-9.1	-8.8	-8.7	-9.0	-8.5	-8.1	-9.1	-8.2	-7.5	-7.4	-11.7	-11.6	-11.4	-11.4
	BECCS Purpose-grown CCS Land	-11.6	-11.6	-11.7	-11.8	-11.5	-11.4	-11.3	-11.7	-11.4	-11.2	-11.1	-12.9	-12.9	-13.0	-13.0
	LULUCF vs. Forest	-21.0	-29.7	-7.3	-10.4	39.8	-3.0	-4.2	-41.9	-3.7	-10.8	-11.2	-17.6	-25.6	-7.5	-9.8
	LULUCF vs. Forest & Other natural	-11.4	-11.6	-8.7	-9.6	-12.3	-7.3	-7.8	-12.9	-9.6	-9.9	-9.6	-9.9	-10.2	-7.9	-8.8
Key metrics	Purpose-grown Yield (GJ per ha)	204	208	213	215	206	208	207	199	199	198	199	203	208	215	218
	Residue & MSW share in biogenic carbon	37%	38%	42%	45%	38%	40%	43%	56%	60%	65%	67%	37%	38%	41%	44%
	CCS sector share	75%	73%	69%	67%	73%	68%	65%	72%	68%	63%	61%	88%	86%	84%	82%
	CCS sector share Purpose-grown	80%	78%	75%	74%	78%	74%	72%	77%	72%	67%	66%	91%	89%	88%	87%
	CCS sector share Residue & MSW	67%	65%	61%	59%	65%	60%	57%	68%	65%	61%	59%	83%	81%	78%	76%
	Capture rate in CCS sectors	68%	67%	66%	66%	67%	66%	65%	71%	70%	69%	68%	76%	74%	72%	72%
	Share of biogenic carbon captured	51%	49%	46%	44%	49%	45%	43%	51%	47%	43%	42%	66%	64%	61%	59%

#	Reviewer #2 Remarks (1 st Round)	Responses/Actions (1 st Round)
B0	This paper provides noteworthy results by homogenizing mitigation policies for reforestation/afforestation and BECCS strategies, and comparing effects on land use, LULUCF, food prices... Minor revisions are suggested below.	Thank you for the careful reading of our paper and the valuable comments. We believe the paper has been significantly improved as a result of the very useful comments and suggestions. Please note that in the response/revision:  • Comments/responses are indexed in case of future communications, e.g., B1. • Changes made are highlighted in both the Main (track-change) manuscript and SI (redline). • SI figures are reindexed since new figures were added. Detailed point-by-point responses are provided below.
B1	Abstract: Define / spell out GCAM and IAM	The abstract has been rewritten and shortened now due to the 150-word limit. However, it seems model (e.g., widely-used IAMs) names were not written out in recent papers published in nature journals.
B2	Figure 1: the color choice makes it difficult to distinguish the 2C main scenario and the 2C A/R focused scenario.	We updated the color palettes and improved the annotations (see Fig. RR6 or the updated Fig. 1). Please also note that the Main and A/R-Focused scenarios are identical under no land mitigation policy (No-LCP), so the same color is used for the two scenarios in the legend table. In addition, we include the source data for generating the figure, per journal recommendations. The code/workflow for generating the figure is also provided in the GitHub repo (realxinzhao/paper-nc2024-LandBasedCDR-DisplayItems). We believe the higher resolution version of the figure will be used in publication.
B3	Line 52: the authors point out the statistically negative relationship between the cumulative removals from BECCS and LULUCF. Are these two CDR methods also competing with land for food production? Or other types of land use (like pristine land)?	Yes. Land competition is modeled for all land in our model, GCAM. For example, in Fig. 3D, we showed the land use change results, indicating the competition among Cropland: Energy (purpose-grown energy crop), Cropland: NonEnergy (e.g., for food, fiber, or other uses), Forest, Pasture, and other natural land (i.e., grassland and shrubland). Land coverage, definition, and modeling could be somewhat different across different IAMs. But GCAM includes all land covers and employs a nested logit approach to model their competition. A detailed description was provided in Section 2.2. But to clarify this, we now include the following sentence in the Method section (GCAM description): “GCAM includes all land covers and employs a nested logit approach to model their competition (Section S2.2).”
B4	Line 72-76: this sentence seems counter intuitive and might require additional explanations. Does it mean that maintaining and preserving our forests would not be as efficient than having energy crops for BECCS, from a carbon capture and storage viewpoint?	Thanks. The sentence (originally lines 72 – 76) “On the other hand, more realistic forest protection or A/R policies.....limiting the amount of mitigation that can be delivered” does not compare BECCS and A/R. We mean to highlight the uncertainty of carbon removal caused by the potential ineffectiveness of land-system policies (e.g., monitoring, reporting, verification, permanence, and other

#	Reviewer #2 Remarks (1 st Round)	Responses/Actions (1 st Round)
		factors.), as discussed in the cited studies in the paragraph ²⁻⁴ and in Badgley et al. (2022)⁵ (California Air Resources Board’s example). As described later in Section 1, we added partial land carbon pricing scenarios to address the uncertainty surrounding land-system policies. We have revised this sentence to clarify this (along with a few other updates around the same paragraph): “In practice, due to sustainability concerns⁶, potential biophysical impacts (e.g., albedo and evapotranspiration)⁷, and institutional challenges (e.g., measurement, reporting, verification, and permanence protocols)^{4,5,8,9}, the strength, scope, and effectiveness of land-system mitigation policies could be highly uncertain and, thus, deserves careful consideration by decision-makers. Compared to the UCT regime, a full land-system mitigation policy, forest protection or A/R policies, e.g., the expanded Reducing Emissions from Deforestation and Forest Degradation (REDD+) framework or zero-deforestation supply chain policies, may represent land-system mitigation policies with only partial strength and/or partial land coverage, potentially limiting the amount of mitigation that can be delivered^{2,3}.”
B5	Line 78: how do monitoring, reporting, and verification protocols vary? How do monitoring, reporting, and verification differ between various CDR methods?	Monitoring, reporting, and verification (MRV) are not explicitly considered in our study. This was clarified in the Discussion section: “Our scenarios assume no risk of unplanned reversal of carbon storage on land, but in the real world, the viability and efficiency of A/R depend on how effectively carbon is stored over time, which in turn depends on the quality of monitoring, reporting, and verification (MRV) protocols.” Please also see clarifications in B4 above.
B6	Line 85: could you give a couple of examples to understand better what is behind “land” and “energy carbon”. Does land refer to A/R, and energy carbon refer to crops for BECCS?	The original sentence: “100%-LCP scenario applies the same price to land and energy carbon.” Land carbon means land carbon storage, and changes in land carbon correspond to LULUCF emissions. Energy carbon here was meant to be Fossil Fuels and Industry (FFI) emissions. Note that total carbon emissions = LULUCF + EIP (Energy and Industrial Processes). And EIP = FFI + BECCS in our study (clarified in Methods). The sentence has now been updated: “The No-LCP scenario assumes no land mitigation policy, whereas the 100%-LCP scenario applies the same price to land carbon storage and FFI carbon.”
B7	Line 90: could you explain in greater details how the land carbon pricing works?  What is the cost of A/R and of BECCS? What is the value given to the land carbon pricing? Would the land carbon pricing fully pay for the deployment of these technologies? Could it be a source of revenue for land owners? Would a land carbon pricing that is too high compete with food production? 	Thank you for the detailed questions. It’s worth noting that our focus in the main text was on highlighting advancements, such as sectoral policy differentiation in a harmonized framework and the decomposition of land carbon removal efficiency. Many of our modeling methods were established previously, so, where applicable, we included references and provided comprehensive documentation in SI. We believe that the reviewer has likely found answers to most of these questions in the SI, especially in Section S2.4 for land carbon pricing methods, and in the Methods section, as suggested by subsequent comments. Here, we offer brief responses and direct reviewers to the Methods and SI for more detailed information.  From the perspective of landowners, they make decisions on land allocation, i.e., whether to use the land for A/R, dedicated energy crops, or others, based

#	Reviewer #2 Remarks (1 st Round)	Responses/Actions (1 st Round)
		on their expected rental profits. Many factors will affect the rental profits (Sections S2.2 & S2.3). The cost of A/R is implied by the land conversion costs and cost of BECCS will be the technologies cost difference between CCS and NonCCS in a specific demand sector (Section S2.8).  b. The carbon prices for land are linked to the carbon prices in Fossil Fuels and Industry, which was indeed how our LCP (land carbon pricing) scenarios were designed. See the Method section (Differentiating mitigation efforts by sectors or technologies) & Fig. 1 legend for more details. c. Land carbon pricing perturbs land market equilibrium. The cost of land conversion is implied by the difference in rental profit and landowners's preferences (Section S2.4). d. Yes, landowners receive credits for carbon storage. See Equation S3 in Section S2.4. e. Yes, land carbon pricing indeed lead to stronger land competition and higher agricultural prices (Fig. 6). Please see additional details in the responses to B11 below.
B8	Line 112: this paragraph compares A/R-focused scenarios with the main scenario that has relatively more BECCS. Does this study account for potential displacement of fossil fuels by higher energy production with BECCS (and potentially transforming current fossil fuel power plants in BECCS facilities)? Does it also account for the increasing energy demand?	Here are some explanations and clarifications:  1. Please note that this paragraph is a brief overview and a more detailed discussion of A/R-focused vs. Main is provided in a later Section (Sensitivity of CDR deployment to alternative land and energy system policies). 2. Yes. GCAM energy system includes the technologies and represents the competition, transformation, and substitution among the technologies. E.g., higher BECCS-electricity and lower fossil fuel-electricity over time in mitigation scenarios. And energy demand changes driven by population and income growth and energy system changes have been included. 3. See additional details in Methods, Section S2.8, and Fig. S11. 4. These are also implied by the emission decomposition (Fig. S14) and primary bioenergy supply and demand by sector (Fig. S17).
B9	Line 121: is FFI carbon prices similar to the 45Q tax credit (tax credit in the United States recently increased by the Inflation Reduction Act)?	Here are some explanations and clarifications:  1. FFI carbon prices are likely not quantitatively related to the 45Q tax credit or the US Inflation Reduction Act (IRA). 2. However, they share a theoretical similarity in function since IRA policies may imply an effective carbon rate in energy systems. That is, they both serve to facilitate low-carbon transitions through market-based measures (e.g., perturbing market equilibrium by introducing a wedge, such as taxes, quotas, or credits, between producer and consumer prices). 3. Similarly, our land carbon pricing is theoretically/ qualitatively similar to existing efforts that protect and preserve forests and natural areas, e.g., REDD+, Bonn Challenge, etc. 4. Though our study does not focus on specific sectoral mitigation policies like IRA, a recent study by Bistline et al. (2023)¹⁰ utilized a version of GCAM to

#	Reviewer #2 Remarks (1 st Round)	Responses/Actions (1 st Round)
		examine the impact of the US IRA.
B10	Line 137: Do the forestry residue come from A/R? Are the A/R scenario combined with the BECCS scenario with for instance forest thinning for fire mitigation?	Good questions. In our modeling, forestry residue does come from the forestry sector which uses managed forest land to produce forestry products, but not unmanaged forest land. In Fig. RR7, we highlight the land use change results for managed and unmanaged forests. Managed forest land, associated with forestry product demand and economic growth (with price and income elasticities), exhibits relatively greater stability compared to unmanaged forest land (with relatively higher land carbon density), which is relatively more responsive to land carbon policies. The breakdown of residual biomass by source (crop vs. forest) is depicted in Fig. RR8, confirming that residual biomass from forestry is less sensitive to land carbon policies in comparison to crops. It is important to note that while land carbon policies result in avoided deforestation and a consequent reduction in fires for forest clearing (leading to moderately lower NonCO2 emissions, as indicated in Fig. RR9), our results and modeling do not demonstrate a significant synergy between forest-residue-based Bioenergy with Carbon Capture and Storage (BECCS) and the reduction of natural forest fires. A more in-depth exploration of this aspect is deferred to future work. The following sentence is added to Section S2.7 (Residual biomass supply) “It is worth noting that the supply curve for forestry residues (both primary and secondary) is tied to forestry products which are produced using managed forest land, not unmanaged forest land.” The following sentences are added to Section S3.2 “In addition, residual biomass from both crop and forestry does not vary significantly across scenarios as they are constrained by the supply of crop and forestry products (Table S5). In particular, forest products are produced from managed forest land. And managed forest land, associated with forestry product demand and economic growth (with price and income elasticities), exhibits relatively greater stability compared to unmanaged forest land (with relatively higher land carbon density), which is relatively more responsive to land carbon policies.” In addition, Tabs RR1 (Table S5 in SI) is also added to provide source data for figures and support the scenario comparisons.
B11	Line 151: if A/R and BECCS lead to higher agricultural prices, is there a risk of food shortage? The need for food will necessarily conflict with climate goals. If there is a risk of food shortage, how realistic are these scenarios?	Food shortage should probably be measured in relative terms. It is certain that when agricultural & food prices are higher (e.g., driven by mitigation policies) compared to business-as-usual (the reference scenario), consumption would be lower (implying a negative price elasticity of consumption). However, market-mediated results like yield intensifications and demand-side adaptations (e.g., reduced crop consumption for feed and other purposes) will alleviate the impacts.

#	Reviewer #2 Remarks (1 st Round)	Responses/Actions (1 st Round)
		This was discussed briefly in the main Section (Agricultural price implications of land-based mitigation measures) with additional descriptions in Section S3.2 and Fig. S35. In addition, please also note that in both reference and the mitigation scenarios, per capita dietary energy available is increasing overtime. We also provide additional results showing this (Fig. RR9 and new Fig. S36) to ensure no unrealistic “food shortage” under our mitigation scenarios. The following description is added to Section S3.2. “The world average dietary energy available is about 2960 kilocalories per capita per day (kcal/ca/d) in 2020, and it is projected to grow by 27% to 3770 kcal/ca/d by the end of the century in the reference scenario (Fig. S38). Under the mitigation runs, the dietary energy supply could decrease moderately by 12 (Low-Bioenergy & No-LCP) to 85 (Low-Bioenergy & 100%-LCP) kcal/ca/d compared to the reference projection. Regional impacts could be larger, particularly for regions that are more sensitive to land-based mitigation policies, e.g., AFRICA (-105 kcal/ca/d in Low-Bioenergy & 100%-LCP and -90 kcal/ca/d in 1.5C & 100%-LCP) and LATIN_AM (-105 kcal/ca/d in Low-Bioenergy & 100%-LCP).”
	Also, do these estimates consider a growing global population?	Yes, our estimates considered growing populations along with GDP and productivity changes, and they are key drivers in all the scenarios. Please find details in Section S2.1 and Fig. S5 .
B12	Line 156: “In the reference scenario, global non-energy cropland expands over time”, this means that non-energy cropland is taking over some other land uses, which type of lands are these? Land for food production? Forests? Other natural habitats?	Here are some explanations and clarifications:  1. Yes, non-energy cropland (e.g., mainly food crops) is expected to grow in the reference scenario, mainly driven by population and income growth. The land use change results in the reference scenario are provided in Fig. S12C. As explained in Section S3.2, pasture and other natural land (grassland and shrubland) are the major land sources. 2. Please note that reference scenario results are mostly provided in SI Section S3.2. We intended to use non-energy cropland expansion in the reference projection to explain the same but weakened trend in the 2C No-LCP scenario, and then compare it with other scenarios with land carbon pricing (10%LCP-100% LCP) to explain the reversed trend (decreasing non-energy cropland). 3. We now rewrite the sentence to include more details: “Global non-energy cropland expands over time in the reference scenario (Section S3.1), but this trend is weakened when EIP carbon is priced in the 2 °C No-LCP scenario as the demand for energy crops is higher and reversed with stronger land carbon policies as land competition further intensifies due to the higher demand for A/R and natural land (Fig. 3D).”
B13	Line 158: “Global net deforestation switches to net afforestation”. The “net” metric might need to be discussed, as afforestation is difficult to achieve with	The “net” here simply means global net total, i.e., the sum total of forest area changes across regions. That is, it does not mean a net of failed afforestation

#	Reviewer #2 Remarks (1 st Round)	Responses/Actions (1 st Round)
	success. Numerous afforestation projects have seen low success rate with a high mortality of trees, and little oversight, and not necessarily prioritizing native species. Please provide more details on how reforestation and afforestation plan to be deployed.	projects. The revision of the first sentence (B12) provides a context of this description, i.e., results in Fig. 3D. So we simply delete “net” in the sentence to avoid confusions: “Global deforestation switches to afforestation with a land policy in-between 10%-LCP and 50%-LCP.” In addition, as discussed in B10, our land carbon pricing is theoretically/qualitatively similar to existing efforts that protect and preserve forests and natural areas. In the modeling, land carbon pricing affects land market equilibrium and encourages transitions from low-carbon land to high-carbon land. In the modeling, land carbon policy is linked (with different strengths to reflect uncertainty) to energy system carbon policy.
B14	Line 192: what makes pasture available for conversion? Is that a type of land projected to be less needed for livestock in the future?	Here are some explanations and clarifications:  1. When comparing A/R-focused (pasture and natural land won't receive credit for land carbon storage) vs. the 2C Main scenario (all land receive credit for land carbon storage), i.e., line 192 (original), more pasture will be converted for A/R to receive the higher land carbon credit. This was indeed a result of profit-seeking landowners' responses to land carbon policies. 2. However, even under 2C Main or the reference scenario, pasture is still the largest land source (in case this is also part of the question). E.g., in the reference scenario, total pasture land decreases as cropland expands. This is consistent with recent observations (see Taylor and Rising, 2021)¹¹ that global pasture land expansion has plateaued in the past few decades and started to decrease, driven by the intensification of the livestock sector. 3. Thus, yes, pasture is “a type of land projected to be less needed for livestock in the future”, but “less needed” means relatively smaller rental profits (shadow land prices) compared to other uses, so it is converted and land carbon policies further encouraged such conversion.
B15	Line 269-270: this is a major conclusion, encouraging efforts from all sectors and the deployment of a portfolio of solution as also suggested by your statement lines 278-280.	Indeed. Thanks. We also expanded the Discussion sector to highlight the key insights from the results.
B16	Line 329: could natural land with relatively high carbon density be protected, to improve ecosystems conservation and reduce the risk of conversion?	Good question. Here are some explanations and clarifications:  1. As explained in Section S3.5, protecting land for carbon storage and our land carbon pricing (carbon subsidy) approach could be theoretically and functionally equivalent. In other words, our land carbon pricing is a sort of “land carbon protection”. Technically speaking, land carbon pricing is more general as land protection will have concerns about property rights. For example, the US Conservation Reserve Program (CRP) cannot force farmers to remove (protect) environmentally sensitive land from agricultural production as farmers have the right. Instead, CRP utilizes the land market to provide a rental

#	Reviewer #2 Remarks (1 st Round)	Responses/Actions (1 st Round)
		payment to farmers to “protect” the land.  2. In our framework, land with higher carbon density indeed receives higher carbon credit (see Section S2.2). 3. Please also note that about 30% of the natural land is “protected” (fixed) in our modeling, and that includes land currently under conservation policies (see Section S2.2).
B17	Line 376-377: this sentence might have to be revised. “derived” or “based on” should be removed.	Done. Thanks.

■ AR6 CB (Gt CO₂) ● This study (GCAM)

Land carbon pricing (LCP) strength

EIP and land system mitigation policy

2 °C (Main) Global EIP system-wide carbon pricing with an end-of-century CB of 1150 GtCO₂; credit all land for carbon storage.

2 °C A/R-Focused Credit only forest land for carbon storage.

2 °C Low-Bioenergy Limit primary bioenergy by a linear path to 100 EJ in 2100.

1.5 °C End-of-century CB of 500 GtCO₂.

Land carbon is priced as a portion of carbon prices in Fossil Fuels and Industry, i.e., 0% (No-LCP), 10%, 50%, or 100% (100%-LCP).

Fig. RR6 (Fig 1 in the main manuscript) Contributions of land-based carbon dioxide removal measures. The left panel shows the relationship between global cumulative CO₂ removals/emissions in 2020 – 2100 for LULUCF and BECCS, while the right panel shows the relationship between LULUCF and Energy and Industrial Process (EIP) emissions, projected by climate change mitigation pathways. Each dot represents a projection from an IPCC AR6 pathway (square) or a mitigation pathway generated in the present study using GCAM (round). The square dots (n = 604) in both panels are projections from IPCC AR6 1.5 °C and 2 °C pathways and with Carbon Budgets (CBs) in [175, 1475] GtCO₂, distinguished by CB subranges (filled color). The boxplots on the sides show the median values (line), the 1st and 3rd quartiles (boxes), and the 5 – 95 percentile ranges (whiskers) of the AR6 pathways; the blue line on the boxplots shows the median value of the full range. The blue dotted lines in the main panels are fitted using quantile regression at the 5th, 50th, and 95th percentiles. The round dots (n = 15) represent GCAM projections, with scenarios distinguished by filled colors and described in the legend. Note that Main and A/R-Focused scenarios are identical under no land mitigation policy (No-LCP). Points on a diagonal line in a panel have the same total removals/emissions, i.e., land-based carbon removals (left) or CBs (right). The beta coefficient between LULUCF and BECCS, with CB controlled, stands at -0.17 for AR6 pathways. In contrast, the corresponding value for GCAM scenarios studied is -0.86 (-2 for the 2 °C Main scenarios). For more details about AR6 pathways, see Section S1. Data source: AR6 Scenario Database and GCAM simulation results.

Fig. RR7 (Not included in the manuscript since the information has been included in Fig.S23) | Global land use and land use change for managed and unmanaged forests relative to 2020. The annual mean value is added in the last column in each panel).

Fig. RR8 (Not included in the manuscript since the data are in Table S5) | Projections of global 2020 – 2100 cumulative primary residual bioenergy supply by source (crop and forestry)

Fig. RR9 (Not included in the manuscript since the information has been included in Fig.S16) | Projections of global 2020 – 2100 cumulative non-carbon dioxide GHG emissions from unmanaged land. Note that unmanaged land emissions were from natural forest and grassland fire.

Fig. RR10 (New Fig. S36) | Projections of dietary energy availability at the world (Panel A) and regional (Panel B) levels. The black lines show the reference projections.

#	Reviewer #3 Remarks (1 st Round)	Responses/Actions (1 st Round)
C1	The paper presents a comprehensive study using the Global Change Analysis Model (GCAM) to investigate the trade-offs between two major land-based carbon dioxide removal (CDR) strategies: afforestation/reforestation (A/R) and bioenergy with carbon capture and storage (BECCS). The study adeptly utilizes various policy scenarios to explore these trade-offs, focusing on aspects such as land competition, carbon pricing, and impacts on agricultural prices. The findings contribute to a deeper understanding of how land-based CDR strategies might perform under different global warming targets and policy frameworks.	Thank you for the careful reading of our paper and the impartial evaluation. Please note that in the response/revision:  • Comments/responses are indexed in case of future communications, e.g., C1. • Changes made are highlighted in both the Main (track-change) manuscript and SI (redline). • SI figures are reindexed since new figures were added. Detailed point-by-point responses are provided below.
C2	Key Findings: Trade-offs between A/R and BECCS: The study identifies a trade-off between A/R and BECCS, as both are land-intensive and compete for the same resource. An inverse relationship exists between their deployment, with a partial valuation of land carbon storage enabling a net land sink even under large-scale BECCS deployment. Impact of Carbon Pricing on Land Use and CDR: Various carbon pricing scenarios show that higher land carbon prices lead to increased land carbon storage and reduced land use for food and energy. Conversely, weaker land mitigation policies result in the opposite effect. This variation significantly affects the land-based CDR strategies deployed. Effects on Agricultural Prices: The scenarios indicate that land-based CDR measures, particularly when associated with strong land carbon pricing, could lead to significant increases in agricultural prices. Demand-side adaptations and yield intensifications could mitigate this increase. There is a complex interaction between land-system mitigation policies, agricultural markets, and food security. Policy-Driven Variations in Land Use and CDR Efficiency: The study explores different policy scenarios, including those that focus exclusively on forest carbon or impose constraints on bioenergy. These scenarios demonstrate how policy choices can significantly affect land use patterns and the efficiency of land-based CDR measures. Land Carbon Removal Efficiency: The paper calculates land carbon removal efficiency, considering both BECCS and A/R, and finds significant variations across scenarios. The efficiency of BECCS improves over time due to yield growth and technological progress, whereas the efficiency of A/R tends to decline as forests mature.	Thanks for the detailed summary. In this revision, we have the opportunity to expand the main paper (by about 50%). Thus, we have provided more detailed discussions of the scenarios (Introduction & Results) and key findings (Discussion). We believe the clarity of communication has been significantly enhanced. In addition to the key findings nicely summarized, we also highlight a few more here: Comparison to AR6 pathways: Projections from our scenarios exhibited a reasonable range of variation compared to the results from AR6 pathways regarding land-based removals, energy system carbon prices, land removal density, and agricultural prices. It is plausible that harmonizing land-based mitigation policies and related assumptions among IAMs could enhance the level of agreement in their projections. Extensive margin responses: A/R removal intensity diminishes when forest expands, especially under increased land competition. This extensive margin response stems from Ricardo’s Law of Rent, which states that the most productive land is used first, so that, all else being equal, marginal expansion into lower-productivity land drives down the mean productivity¹². While purpose-grown energy crops may also have extensive margin responses when production expands, the yield for purpose-grown energy crops is presumed to increase over time, with price-induced yield intensification also contributing¹³. Potential tradeoffs between removal effectiveness and agricultural price responses. We find a generally positive relationship between agricultural prices and the land removal intensity of LULUCF (e.g., A/R is more effective when expanding into low-carbon land, i.e., cropland), given that both are sensitive to cropland conversion, suggesting that such trade-offs may be difficult to avoid completely in land system mitigation. A few points related to the land removal intensity/efficiency are further elaborated in more detail (please see the Discussion section).
C3	Strengths:	Thanks.

#	Reviewer #3 Remarks (1 st Round)	Responses/Actions (1 st Round)
	 • The paper effectively utilizes GCAM to explore complex interactions between land-based CDR measures and policy scenarios. • It comprehensively analyzes the trade-offs between A/R and BECCS, highlighting critical aspects like land competition, carbon pricing, and agricultural prices. • The study contributes valuable insights into policy-driven variations in land use efficiency and their implications for global carbon budgets and agricultural markets. 	
C4	Weaknesses:  • While the study accounts for various policy scenarios, it might not fully capture the complexities of real-world policy implementation and socio-economic dynamics. 	This is true. However, technically speaking, no model “fully capture[s] the complexities of real-world policy implementation and socio-economic dynamics”. In addition, please note that our focus is on scenario comparison in an internally consistent framework and on understanding the sensitivity, as opposed to a single scenario projection.
	 • The paper focuses on the efficiency of land-based CDR methods but might not sufficiently address these strategies' potential environmental and social impacts. 	Thanks. Our study enhances the understanding of the role of land-based CDR. The explored results were predominantly first-order (from GCAM), emphasizing market-mediated responses and related environmental implications (e.g., carbon budget overshooting, NonCO2 emissions, land allocation, etc.). While broader studies on corresponding environmental and social impacts could be valuable, a comprehensive understanding of the first-order results remains crucial and foundational. For instance, a recent study by Huang et al. (2023)¹⁴ investigated air quality and health impacts of mitigation policies, generating 30,000 scenarios using a coupled modeling framework. However, land-system mitigation policies were overlooked, and all scenarios were deforestation scenarios. Similarly, biodiversity studies¹⁵ have relied on IAMs' land use projections, sensitive to the strength and scope of land-based mitigation policies (as demonstrated in our work). Although our focus isn't comprehensively assessing environmental and social impacts, our results significantly influence past studies in this domain. We believe our study establishes a robust foundation for further exploration of broader environmental and social impacts (see the Discussion section).
	 • Though robust, reliance on a single modelling framework might limit the generalizability of the findings, as different models might yield different outcomes under similar scenarios. 	We indeed approach this point from a different perspective. We have observed variations in model outcomes regarding land-based mitigation measures, and these model differences are not well understood. This gap in the literature motivated us to conduct this study. Our scenarios successfully represented the variations in BECCS and LULUCF seen in AR6 pathways, and we believe our results contribute to the existing literature. We have clarified this point in both the introduction and discussion sections. Regarding the "generalizability" of our findings, we recognize that opinions may differ. We have summarized the key findings in the Discussion section and firmly believe that these findings are "generalizable," notwithstanding potential parameter

#	Reviewer #3 Remarks (1 st Round)	Responses/Actions (1 st Round)
		uncertainty. Further discussions on model intercomparison are provided below.
C5	Comparison with Existing Literature: The study aligns with the current discourse in the field, particularly regarding the trade-offs and interactions between A/R and BECCS strategies. Its focus on policy-driven variations in land use and CDR efficiency adds valuable insights to existing literature, especially regarding the potential impacts on agricultural prices and land use patterns. However, while the study extends the discussion on land-based CDR strategies, it does not significantly diverge from existing approaches or introduce novel methodologies that would distinguish it substantially from previous research. The manuscript's findings, while insightful, align closely with established understandings in the field, as outlined in the Intergovernmental Panel on Climate Change (IPCC) reports and other comprehensive studies.	Thanks. As discussed in C2 above, we now include a more detailed discussion of the key findings and insights, most of which were not well understood by the literature. Please see C7 below on the methodological advancements. Additionally, in the revision, we also clarified and highlighted our comparison with the AR6 pathways (as also suggested by Reviewer 1).
C6	Recommendations: Broader Contextualization: To enhance the manuscript's appeal, it could benefit from a broader contextualization of its findings within the global climate change mitigation landscape. This might include comparing these land-based strategies with other mitigation options or considering policy choices' socio-economic and environmental implications more comprehensively.	Thanks for the suggestions. The paper has been expanded to include a more detailed discussion of scenarios and key findings, leading to an improved contextualization, with a continued focus on the role of BECCS and land-system mitigation policies. This expansion particularly enhances communication with stakeholders involved in BECCS and land-system mitigation policies, as well as those developing IAM models and generating new scenarios. Please refer to C4 above for discussions on socio-economic and environmental implications. The scenarios designed in our study were not intended for a direct comparison across CDRs. Additionally, a recent study by Fuhrman et al. (2023)¹⁶ utilized GCAM to assess six CDRs in deep mitigation scenarios, and it could be further extended in future work to consider the sensitivity of comprehensive land-system policies implied by our study.
C7	Comparing methodologies: This is a big ask, but incorporating or comparing results from different models or methodologies could provide a more robust and diversified analysis, strengthening the manuscript's contribution to the field. By comparing and contrasting results from these models with those from GCAM, researchers can gain insights into the robustness of their findings and explore a broader range of scenarios and policy implications. Each model has its own strengths and limitations, and by using them in conjunction, it's possible to provide a more nuanced understanding of the complex interactions between human activities, energy systems, and the environment. For instance, while GCAM provides a detailed representation of the energy and land-use sectors, MESSAGE or WITCH might offer different perspectives on technological change and innovation in the energy sector. Similarly, comparing the socioeconomic implications of climate policies using both GCAM and REMIND or AIM/CGE	Thanks for the kind suggestion of a multi-model approach. We agree with the reviewer that a multi-model approach is useful for understanding model differences. As modelers, we appreciate the past model intercomparison projects (GCAM has participated in quite a few of them). Relevant recent model intercomparison studies include Roe et al. (2021)¹⁷, Fujimori et al. (2022)⁶ (AgMIP), and Hasegawa et al. (2021)¹⁸ (ENGAGE), as they have explored model differences focusing on the land-based mitigation results. However, in our opinion, it is not necessarily true that a multi-model approach is required or that it enhances novelty for publications in high-impact journals. In many cases, a multi-model approach could be less useful than it seems to be. Because (1) Please note that model differences do not arise from a common data-generating process, making it challenging to draw meaningful statistical

#	Reviewer #3 Remarks (1 st Round)	Responses/Actions (1 st Round)
	could yield a more comprehensive view of the potential impacts on different regions and economic sectors.	inferences from results across different models. That is, distributions of outcomes from various models may not indicate statistical robustness. (2) There is a lack of theoretical basis for the multi-model approach in terms of comparability across models. Due to differences in data and parameters, model results are expected to be different. Thus, more technical comparisons of models, considering their structure, data, and assumptions of parameters, could be more useful than a simple results comparison (E.g., see recent studies^{12,19–21} explaining how harmonizing data, approaches, & parameters may lead to harmonized results from economic equilibrium models). However, the community has not invested adequate efforts to harmonize data, modeling structure, and parameter differences—possibly because a cheaper model intercomparison of results can be published in high-impact journals without addressing the underlying issues. More specifically, in our study: (1) We agree that “each model has its own strengths and limitations”. We believe GCAM has strengths in studying land-system mitigation policies in an internally consistent framework. In a sense, our scenario design is novel, and (most if not all) other models have not and may not be able to test those scenarios. Particularly, our study generates pathways with differentiated mitigation efforts, including partial land carbon pricing (i.e., pricing land-system emissions differently from energy and industrial process emissions) and asymmetric sectoral carbon pricing (i.e., pricing forest land carbon differently from other land or pricing BECCS differently from other energy sector emissions). By distinguishing mitigation efforts by sector, our analysis provides novel insights into sectoral shadow prices of carbon and their impact on agriculture and energy markets and land carbon removal intensity. Such an analysis has never been performed before, to our knowledge. (2) Our calculation of land carbon removal intensity represents an advancement compared to the “mitigation density” used in Roe et al. (2021) (see discussions in Section S3.3). Additionally, our decomposition of the removal intensity is novel and offers new insights. For example, (1) our results illustrate that not all BECCS are equally land-intensive, and not all energy crops are used in combination with CCS. This distinction is crucial but was overlooked in the prior multi-model studies^{6,17,18} mentioned earlier. (2) We show that A/R removal intensity is sensitive to the study periods and subject to extensive margin responses. I.e., A/R removal intensity diminishes in the long run (as the forest matures) and across space when the forest expands (Ricardo’s Law of Rent applies). These aspects were also not addressed in previous studies, with some studies^{6,17} focusing on projections only until 2050, potentially missing the full benefits of A/R, and uncertainty regarding how A/R (or land-system mitigation) is represented in different models. (3) Our study indeed speaks to researchers and IAM models in the community, as how and to what extent land-system mitigation policies are implemented have

#	Reviewer #3 Remarks (1 st Round)	Responses/Actions (1 st Round)
		been recognized as areas of substantial disagreement in integrated assessment modeling. Projections from our scenarios exhibited a reasonable range of variation, compared to the results from AR6 pathways, regarding land-based removals, energy system carbon prices, land removal density, and agricultural prices. It is plausible that harmonizing land-based mitigation policies and related assumptions among IAMs could enhance the level of agreement in their projections. Our study establishes a foundation for further assessing the impacts of carbon mitigation measures and exploring the implications of trade-offs, considering more comprehensive environmental consequences in model intercomparison frameworks. Thus, we believe our approach represents a significant advance. In the revision, the contributions and key findings are highlighted.
C8	Suitability for Nature Communications: While the research is methodologically sound and the findings are indeed valuable, the scope of the study, focusing primarily on the trade-offs between two specific land-based CDR strategies under a limited set of policy scenarios, may be too narrow for a broad-based, multidisciplinary journal like Nature Communications. Nature Communications often seeks research that has wide-ranging implications, novel approaches, or ground-breaking findings that can appeal to a broad scientific audience. In this context, while rigorous and well-executed, the manuscript might not meet the criteria for a substantial contribution to the field in a way that would justify publication in such a high-impact journal. Given the manuscript's current scope and focus, it might find a more suitable audience in a journal specializing in environmental science, climate policy, or land use studies. Such a platform could provide a more targeted audience that would appreciate the specific nuances and contributions of the research.	Thank you once again for your thorough review of the paper and impartial assessments. As discussed above and given the changes made in this revision, we believe our study has wide-ranging implications, novel approaches, or ground-breaking findings that can appeal to a broad scientific audience.

References

1. Hanssen, S. V. *et al.* The climate change mitigation potential of bioenergy with carbon capture and storage. *Nat. Clim. Chang.* **10**, 1023–1029 (2020).
2. Popp, A. *et al.* Land-use protection for climate change mitigation. *Nature Clim Change* **4**, 1095–1098 (2014).
3. Villoria, N., Garrett, R., Gollnow, F. & Carlson, K. Leakage does not fully offset soy supply-chain efforts to reduce deforestation in Brazil. *Nat Commun* **13**, 5476 (2022).
4. Mignone, B. K., Hurteau, M. D., Chen, Y. & Sohngen, B. Carbon offsets, reversal risk and US climate policy. *Carbon Balance and Management* **4**, 3 (2009).
5. Badgley, G. *et al.* Systematic over-crediting in California's forest carbon offsets program. *Global Change Biology* **28**, 1433–1445 (2022).
6. Fujimori, S. *et al.* Land-based climate change mitigation measures can affect agricultural markets and food security. *Nat Food* **3**, 110–121 (2022).
7. Windisch, M. G., Davin, E. L. & Seneviratne, S. I. Prioritizing forestation based on biogeochemical and local biogeophysical impacts. *Nat. Clim. Chang.* **11**, 867–871 (2021).
8. Erbaugh, J. T. Impermanence and failure: The legacy of conservation-based payments in Sumatra, Indonesia. *Environ. Res. Lett.* (2022) doi:10.1088/1748-9326/ac6437.
9. Grassi, G. *et al.* Critical adjustment of land mitigation pathways for assessing countries' climate progress. *Nat. Clim. Chang.* **11**, 425–434 (2021).
10. Bistline, J. *et al.* Emissions and energy impacts of the Inflation Reduction Act. *Science* **380**, 1324–1327 (2023).
11. Taylor, C. A. & Rising, J. Tipping point dynamics in global land use. *Environ. Res. Lett.* **16**, 125012 (2021).
12. Zhao, X., Calvin, K. V. & Wise, M. A. The critical role of conversion cost and comparative advantage in modeling agricultural land use change. *Clim. Change Econ.* **11**, 2050004 (2020).
13. Keeney, R. & Hertel, T. W. The Indirect Land Use Impacts of United States Biofuel Policies: The Importance of Acreage, Yield, and Bilateral Trade Responses. *American Journal of Agricultural Economics* **91**, 895–909 (2009).
14. Huang, X., Srikrishnan, V., Lamontagne, J., Keller, K. & Peng, W. Effects of global climate mitigation on regional air quality and health. *Nat Sustain* 1–13 (2023) doi:10.1038/s41893-023-01133-5.
15. Leclère, D. *et al.* Bending the curve of terrestrial biodiversity needs an integrated strategy. *Nature* **585**, 551–556 (2020).
16. Fuhrman, J. *et al.* Diverse carbon dioxide removal approaches could reduce impacts on the energy–water–land system. *Nat. Clim. Chang.* 1–10 (2023) doi:10.1038/s41558-023-01604-9.
17. Roe, S. *et al.* Land-based measures to mitigate climate change: Potential and feasibility by country. *Global Change Biology* **27**, 6025–6058 (2021).
18. Hasegawa, T. *et al.* Land-based implications of early climate actions without global net-negative emissions. *Nat Sustain* 1–8 (2021) doi:10.1038/s41893-021-00772-w.
19. Zhao, X. *et al.* The impact of agricultural trade approaches on global economic modeling. *Global Environmental Change* **73**, 102413 (2022).
20. Mignone, B. K., Huster, J. E., Torkamani, S., O'Rourke, P. & Wise, M. Changes in Global Land Use and CO₂ Emissions from US Bioethanol Production: What Drives Differences in Estimates between Corn and Cellulosic Ethanol? *Clim. Change Econ.* 2250008 (2022) doi:10.1142/S2010007822500087.
21. Zhao, X., van der Mensbrugge, D. Y., Keeney, R. M. & Tyner, W. E. Improving the way land use change is handled in economic models. *Economic Modelling* **84**, 13–26 (2020).

REVIEWERS' COMMENTS

Reviewer #1 (Remarks to the Author):

With apologies for the delay in submitting my comments, I'm pleased to report that I find the authors' revisions and responses satisfactory. The paper is now much clearer, and the improved points for policy makers in the Discussion represent a huge improvement. I recommend publication.

My one quibble, about which I recommend doing nothing, concerns the issue of including non-CCS-destined biomass in the land efficiency metric. I find the authors' responses illuminating about their motivations, and I like their approach of maintaining their original analysis while also offering a decomposition into BECCS/noCCS. It achieves their original goal and satisfies those, like me, who want something different. I also think the revised approach improves Fig. 3F (the decomposition of land removal intensity) significantly. I take the authors' point about being interested in market equilibrium effects on land-use as being the relevant consideration for the overarching goals of the paper. I guess I'm just stuck on the fact that land removal intensity seems to be about something else. I recommend not making any changes just to satisfy my hang-up about the term. The current approach is very good.

Reviewer #2 (Remarks to the Author):

The reviewer agrees that the manuscript has been significantly improved by the edits made by the authors. Minor revisions are suggested below, mostly to discuss carbon storage further, especially to compare trade-offs of the carbon storage options included within the A/R and BECCS pathways.

The authors only talk about carbon storage in the context of storing carbon in vegetation and soil. However, in the BECCS configuration, CO₂ is stored in underground geologic formations. This might not seem directly related to the present study, but this is a very important difference when looking at the durability and reversibility of carbon storage. The authors acknowledge that their study do not account for potential reversal, but should provide information regarding the durability and reversal potential for carbon storage in vegetation, soil, and geologic formations.

In an A/R configuration, the carbon is by definition captured and stored onsite. This is however not necessarily the case for a BECCS configuration, where CO₂ might have to be transported over long distances to a viable CO₂ storage site. The authors could discuss these differences, and the fact that even if biomass feedstocks are available in a specific region, some locations might not be viable for BECCS due to the lack of nearby CO₂ storage sites. Also, transporting the CO₂ by pipeline will require the build-out of CO₂ pipelines all over the country, and transporting CO₂ by other modes like rail, trucking, or barges will emit CO₂ emissions during transport. Is that taken into account in the present study?

Regarding Figures RR1 and RR2, please re-state in the caption, what "final energy is referring to", and what all the percentages correspond to.

Reviewer #3 (Remarks to the Author):

The authors have fully addressed my original comments and satisfactory defended their fit into NCOMMS.

I recommend this paper be accepted for publication.

#	Reviewer #1 Remarks (2 nd Round)	Responses/Actions (2 nd Round)
A9	With apologies for the delay in submitting my comments, I'm pleased to report that I find the authors' revisions and responses satisfactory. The paper is now much clearer, and the improved points for policy makers in the Discussion represent a huge improvement. I recommend publication. My one quibble, about which I recommend doing nothing, concerns the issue of including non-CCS-destined biomass in the land efficiency metric. I find the authors' responses illuminating about their motivations, and I like their approach of maintaining their original analysis while also offering a decomposition into BECCS/noCCS. It achieves their original goal and satisfies those, like me, who want something different. I also think the revised approach improves Fig. 3F (the decomposition of land removal intensity) significantly. I take the authors' point about being interested in market equilibrium effects on land-use as being the relevant consideration for the overarching goals of the paper. I guess I'm just stuck on the fact that land removal intensity seems to be about something else. I recommend not making any changes just to satisfy my hang-up about the term. The current approach is very good.	Once again, thank you so much for the careful reading of our paper and the valuable comments. The paper has been significantly improved as a result of your very useful comments and suggestions during the entire review process. We consider our removal intensity decompositions as one step in bridging the gap between sectoral analysis and market-equilibrium IAMs. Future studies can extend the framework for more in-depth comparisons. Certainly, more communication and interdisciplinary studies are needed. Thanks!

#	Reviewer #2 Remarks (2 nd Round)	Responses/Actions (2 nd Round)
B18	The reviewer agrees that the manuscript has been significantly improved by the edits made by the authors. Minor revisions are suggested below, mostly to discuss carbon storage further, especially to compare trade-offs of the carbon storage options included within the A/R and BECCS pathways.	Thanks again for the careful reading of our paper and the additional comments. Detailed responses are provided below.
B20	In an A/R configuration, the carbon is by definition captured and stored onsite. This is however not necessarily the case for a BECCS configuration, where CO₂ might have to be transported over long distances to a viable CO₂ storage site. The authors could discuss these differences, and the fact that even if biomass feedstocks are available in a specific region, some locations might not be viable for BECCS due to the lack of nearby CO₂ storage sites. Also, transporting the CO₂ by pipeline will require the build-out of CO₂ pipelines all over the country, and transporting CO₂ by other modes like rail, trucking, or barges will emit CO₂ emissions during transport. Is that taken into account in the present study?	As described in Section S2.8, GCAM incorporates CCS options at the technology levels, illustrating the competition between technologies with and without CCS. The higher cost (levelized) for technologies with CCS was also represented. Our study relies on the default assumptions of cost and removal fractions in GCAM. The non-energy costs, e.g., carbon capture, compression, transport, and storage, are derived based on Dooley and Dahowski (2009)¹. This is now clarified in Section S2.8. Please also note that we do not explicitly model the detailed supply chain of the CCS "service" and only use representative regional suppliers (i.e., with no spatial details but representing the mean costs and technological specifications). This is also clarified in Section S2.8.
B19	The authors only talk about carbon storage in the context of storing carbon in vegetation and soil. However, in the BECCS configuration, CO₂ is stored in underground geologic formations. This might not seem directly related to	Thanks. In GCAM, BECCS uses advanced energy system technologies to convert

#	Reviewer #2 Remarks (2 nd Round)	Responses/Actions (2 nd Round)
	the present study, but this is a very important difference when looking at the durability and reversibility of carbon storage. The authors acknowledge that their study do not account for potential reversal, but should provide information regarding the durability and reversal potential for carbon storage in vegetation, soil, and geologic formations.	lignocellulosic biomass into modern energy carriers while also capturing the biogenic carbon and storing it underground in a geologic formation. The modeling is at the technology levels (please also find the relevant discussion above in B20). The BECCS removal in our work implies the carbon stored in underground geologic formations, which is usually considered “permanent”². Given that durability and reversibility are not focal points in our study/modeling framework, we defer to future studies on these aspects. The following changes have been made to address the comment:  1. We moved the following sentences from Section S3 to Section S2.8 (a more relevant place): “Land-based CDR measures rely on different processes for carbon removal. A/R involves expanding forests to increase carbon sequestration in vegetation and soil. BECCS, on the other hand, utilizes advanced energy system technologies to convert lignocellulosic biomass into modern energy carriers while also capturing the biogenic carbon and storing it underground in a geologic formation².” 2. The following sentences are added to Section S3.5 (Limitations and future work): “Moreover, we did not consider the unplanned reversal of carbon storage in both land (soil and vegetation) and geologic formations (e.g., BECCS). The durability and reversibility differences in land-based mitigation measures³, like potential increased wildfire risks for A/R and the potential for transportation and storage leakage for BECCS, should be addressed in future research.”
B21	Regarding Figures RR1 and RR2, please re-state in the caption, what "final energy is referring to", and what all the percentages correspond to.	The final energy sectors, comprising industry, buildings, and transportation, have been added to the caption of Fig. S19 (Fig. RR1). Please note that the percentages were explained in the caption of Fig. S19 (Fig. RR1), and Fig. S20 includes a reference to that explanation.

#	Reviewer #3 Remarks (2 nd Round)	Responses/Actions (2 nd Round)
C9	The authors have fully addressed my original comments and satisfactory defended their fit into NCOMMS. I recommend this paper be accepted for publication.	Thank you for your time and insightful comments throughout the entire review process!

References

1. Dooley, J. J. & Dahowski, R. T. Large-Scale U.S. Unconventional Fuels Production and the Role of Carbon Dioxide Capture and Storage Technologies in Reducing Their Greenhouse Gas Emissions. *Energy Procedia* **1**, 4225–4232 (2009).
2. Alcalde, J. *et al.* Estimating geological CO₂ storage security to deliver on climate mitigation. *Nat Commun* **9**, 2201 (2018).
3. Chiquier, S., Patrizio, P., Bui, M., Sunny, N. & Dowell, N. M. A comparative analysis of the efficiency, timing, and permanence of CO₂ removal pathways. *Energy & Environmental Science* **15**, 4389–4403 (2022).